


# Alteration of nitrous oxide emissions from floodplain soils by aggregate size, litter accumulation and plant soil interactions

Martin Ley[1,2], Moritz F. Lehmann[2], Pascal A. Niklaus[3], and Jörg Luster[1]

[1]Swiss Federal Institute for Forest, Snow and Landscape Research WSL, Zürcherstrasse 111, 8903 Birmensdorf, Switzerland

[2]Department of Environmental Sciences, University of Basel, Bernoullistrasse 30, 4056 Basel, Switzerland

[3]Department of Evolutionary Biology and Environmental Studies, University of Zürich, Winterthurerstrasse 190, 8057 Zurich, Switzerland

*Correspondence to:* Martin Ley (martin.ley@wsl.ch)

**Abstract.** Semi–terrestrial soils such as floodplain soils are considered potential hotspots of nitrous oxide ($N_2O$) emissions. Microhabitats in the soil, such as within and outside of aggregates, in the detritusphere, and/or in the rhizosphere, are considered to promote and preserve specific redox conditions. Yet, our understanding of the relative effects of such microhabitats and their interactions on $N_2O$ production and consumption in soils is still incomplete. Therefore, we assessed the effect of aggregate size, buried organic matter, and rhizosphere processes on the occurrence of enhanced $N_2O$ emissions under simulated flooding/drying conditions in a mesocosm experiment. We used two model soils with equivalent structure and texture, comprising macroaggregates (4000–250 μm) or microaggregates (< 250 μm) from a N-rich floodplain soil. These model soils were either planted with basket willow (*Salix viminalis* L.), mixed with leaf litter, or left unamended. After 48 hours of flooding, a period of enhanced $N_2O$ emissions occurred in all treatments. The unamended model soils with macroaggregates emitted significantly more $N_2O$ during this period than those with microaggregates. Litter addition modulated the temporal pattern of the $N_2O$ emission, leading to short-term peaks of high $N_2O$ fluxes at the beginning of the period of enhanced $N_2O$ emissions. The presence of *S. viminalis* strongly suppressed the $N_2O$ emission from the macroaggregated model soil, masking any aggregate size effect. Integration of the flux data with data on soil bulk density, moisture, redox potential and soil solution composition suggest that macroaggregates provided more favorable conditions for spatially coupled nitrification–denitrification, which are particularly conducive to net $N_2O$ production, than microaggregates. The local increase in organic carbon in the detritusphere appears to first stimulate $N_2O$ emissions, but ultimately, respiration of the surplus organic matter shifts the system towards redox conditions where $N_2O$ reduction to $N_2$ dominates. Similarly, the low emission rates in the planted soils can be best explained by root exudation of low-molecular weight organic substances supporting complete denitrification in the anoxic zones, but also by the inhibition of denitrification in the zone above, where rhizosphere aeration takes place. Together, our experiments highlight the importance of microhabitat formation in regulating $O_2$ content and the completeness of denitrification in soils during drying after saturation. Moreover, they will help to better predict the conditions under which hotspots and moments of enhanced $N_2O$ emissions are most likely to occur in hydrologically dynamic soil systems like floodplain soils.



## 1. Introduction

Nitrous oxide ($N_2O$) is a potent greenhouse gas with a global warming potential over a 100 year time horizon 298 times higher than the one of carbon dioxide (Forster et al., 2007). Given its role as climate-relevant gas and in the depletion of stratospheric ozone (Ravishankara et al., 2009), the steady increase of its average atmospheric concentration of $0.75 ppb\ yr^{-1}$ (Hartmann et al., 2013) asks for a quantitative understanding of its sources and the factors that control its production. On a global scale, vegetated soils are the main natural terrestrial sources of $N_2O$. Agriculture is the main anthropogenic source and the main driver of increasing atmosphere $N_2O$ concentrations (Ciais et al., 2013).

In soils, several biological nitrogen (N) transformation processes produce $N_2O$ either as a mandatory intermediate or as a by-product (Spott et al., 2011). Under oxic conditions, the most important process is obligate aerobic nitrification that yields $N_2O$ as by-product when hydroxylamine decomposes (Zhu et al., 2013). Under low oxygen ($O_2$) availability, nitrifier denitrification and heterotrophic denitrification with $N_2O$ as intermediate become more relevant (Philippot et al., 2009). At stably anoxic conditions and low concentrations of nitrate, complete denitrification consumes substantial amounts of previously produced $N_2O$ by further reduction to $N_2$ (Baggs, 2008; Vieten et al., 2009). In environments that do not sustain, stable anoxia but undergo sporadic transitions between oxic and anoxic conditions, the activity of certain $N_2O$ reductases can be suppressed by transiently elevated $O_2$ concentration and thus can lead to the accumulation of $N_2O$ (Morley et al., 2008).

Nitrous oxide emissions from soils depend on the availability of carbon (C) and N substrates that fuel the involved microbial processes. On the other hand, given its dependency on $O_2$, $N_2O$ production is also governed by the diffusive supply of $O_2$ through soils. Similarly, soil $N_2O$ emissions are modulated by diffusive $N_2O$ transport from the site of production to the soil surface (e.g. Böttcher et al., 2011; Heincke and Kaupenjohann, 1999). Substrate availability, gas diffusivity, and the distribution of soil organisms are highly heterogeneous in soils at a small scale, with micro-niches in particular within soil aggregates, within the detritusphere, and within the rhizosphere. These can result in "hot spots" with high denitrification activity (Kuzyakov and Blagodatskaya, 2015).

Soil aggregate formation is a key process in building soil structure and pore space. Soil aggregates undergo different stages in their development, depending on the degradability of the main binding agent (Tisdall and Oades, 1982). Initially, highly persistent primary organo–mineral clusters (20–250 µm) are held together by root hairs and hyphae, thus forming macroaggegates (> 250 µm). Upon decomposition of these temporary binding agents and the subsequent disruption of the macroaggregates, microaggregates (< 250 µm) are released (Elliott and Coleman, 1988; Oades, 1984; Six et al., 2004). These consist of clay-encrusted fragments of organic debris coated with polysaccharides and proteins. This multi-stage development leads to a complex relationship between aggregate size, intra-aggregate structure and soil structure (Ball, 2013; Totsche et al., 2017), which influences soil aeration, substrate distribution and pore water dynamics (Six et al., 2004). Often, micro-site heterogeneity increases with aggregate size, thus fostering the simultaneous activity of different $N_2O$ producing microbial communities with distinct functional traits (Bateman and Baggs, 2005). Aggregate size effects on $N_2O$ production and consumption have generally been studied in static batch incubation experiments with a comparatively small number of isolated aggregates of uniform size, at constant levels of water saturation (Diba et al., 2011; Drury et al., 2004; Jahangir et al., 2011; Khalil et al., 2005; Sey et al., 2008), and through modelling approaches (Renault and Stengel, 1994; Stolk et al., 2011). Previous work provided partially inconsistent results,



which led to an ongoing discourse about the interplay of physicochemical properties and different aggregate
sizes in controlling $N_2O$ emission. For example, ostensible inconsistencies may be attributed to the use of
different aggregate size classes, other methodological constraints (water saturation, redox potential), and
differences in microbial communities. The effects of aggregate size, in combination with fluctuating water
saturation, on soil $N_2O$ emissions have, to our knowledge, not been addressed specifically.
Similar to soil aggregates, the detritusphere and the rhizosphere (the zone of the soil that is affected by root
activity (Baggs, 2011; Luster et al., 2009), can be considered biogeochemical hot spots (Kuzyakov and
Blagodatskaya, 2015; Myrold et al., 2011). Here, carbon availability is much higher than in the bulk soil and
thus rarely limiting microbial process rates. The detritusphere consists of dead organic material, which spans a
wide range of recalcitrance to microbial decomposition. Spatially confined accumulations of variably labile soil
litter form microhabitats that are often colonized by highly active microbial communities (Parkin, 1987).
Aggregation of litter particles has been shown to affect $N_2O$ emissions (Loecke and Robertson, 2009). Hill
(2011) identified buried organic-rich litter horizons in a stream riparian zone as hot spots of N cycling. Similarly,
in the rhizosphere, root exudates and exfoliated root cells provide ample degradable organic substrate for soil
microbes (Robertson and Groffman, 2015). Yet, plant growth may also affect soil microbial communities
through competition for water and nutrients (e.g., fixed N) (Bender et al., 2014; Myrold et al., 2011). The
combined effects of these plant-soil interactions on $N_2O$ production have been reviewed by Philippot et al.
(2009). Root-derived bioavailable organic compounds can stimulate heterotrophic microbial activity, specifically
N mineralization and denitrification. Nitrification in turn can be enhanced by the elevated N turnover and
mineralization rates, but may also be negatively affected by specific inhibitors released from the root or through
plant-driven ammonium depletion. The ability of some plants adapted to water-saturated conditions to
„pump" air into the rhizosphere via aerenchyma (gas conductive channels in the root) leads to an improved
oxygenation of the rhizosphere and a stimulation of nitrification (Philippot et al., 2009). Surrounded by
otherwise anoxic sediments, such aerated micro-environments may create optimal conditions for coupled
nitrification–denitrification (Baldwin and Mitchell, 2000; Koschorreck and Darwich, 1998). On the other hand,
transport of $N_2O$ produced in the soil to the atmosphere is may be facilitated via these plant-internal channels,
bypassing diffusive transport barriers and enhancing soil–atmosphere gas fluxes (Jørgensen et al., 2012).
The dynamics of $N_2O$ emissions are strongly coupled to the dynamics of pore water. Re-wetting of previously
dried soil can lead to strong $N_2O$ emissions (Goldberg et al., 2010; Ruser et al., 2006), likely fostered by a
wetting-induced flush in N mineralization (Baldwin and Mitchell, 2000). On the other hand, the drying-phase
after water saturation of sediments and soils can lead to a period of enhanced $N_2O$ emissions (e.g. Baldwin and
Mitchell, 2000; Groffman and Tiedje, 1988; Rabot et al., 2014; Shrestha et al., 2012) when water-filled pore
space (WFPS) exceeds 60% (Beare et al., 2009; Rabot et al., 2014). The increased $N_2O$ production has been
attributed to enhanced coupled nitrification–denitrification (Baldwin and Mitchell, 2000). Depending on the
spatial distribution of water films around soil particles and tortuosity (which is a function of aggregate size and
soil structure), the uneven drying of the soil after full saturation may generate conditions that are conducive to
the formation of anaerobic zones in otherwise oxic environments (Young and Ritz, 2000). Pore water thereby
acts as a diffusion barrier for gas exchange, limiting the $O_2$ availability in the soil pore space (Butterbach-Bahl et
al., 2013). Moreover, pore water serves as a medium for the diffusive dispersal of dissolved C and N substrates,
e.g. from the site of litter decomposition to spatially separated $N_2O$ producing microbial communities (Hu et al.,
2015). Therefore, fluctuations in water saturation efficiently promote the development of hot spots and hot





moments of $N_2O$ emissions in floodplain soils and other semi-terrestrial soils (Hefting et al., 2004; Shrestha et al.,

118  2012).

The main objective of the present experimental study was to contribute to a better understanding of the factors
governing the formation and emission of $N_2O$ in floodplain soils during hot moments after flooding events.
Towards this objective, we performed a mesocosm flooding simulation experiment under controlled conditions,
with model soils of largely similar structure, but differing in the size distribution of original soil aggregates. We
included two additional factorial treatments: a willow-litter addition treatment to assess whether aggregate size
effects are modified by such a detritusphere, and a willow cuttings treatment to test whether aggregate effects
change in the presence of plants, as result of root soil interactions.
We demonstrate that the level of soil aggregation affects $N_2O$ emission rates from floodplain soils through its
modulating control on the model soils physicochemical properties. We further show that these effects are
modified by the presence of a detritusphere and by root–soil interactions, through effects on carbon and N
substrate availability and redox conditions.
**2. Material and methods**
**2.1 Model soils**
In February 2014, material from the uppermost 20 cm of a N-rich gleyic Fluvisol (calcaric, humic siltic) with
20% sand and 18% clay (Samaritani et al., 2011) was collected in the restored Thur River floodplain near
Niederneunforn (NE Switzerland 47°35' N, 8°46' E, 453 m.a.s.l.; MAT 9.1 °C; MAP 1015 mm). After removing
plant residues such as roots, twigs and leaves, the soil was mixed and air-dried to a gravimetric water content of
$24.7 \pm 0.4$ %. In the next step, the floodplain soil material was separated into a macroaggregate fraction (250–
4000 µm) and a microaggregate fraction (< 250 µm) by dry sieving. The threshold of 250 µm between
macroaggregates and microaggregates was chosen based on Tisdall and Oades (1982). Soil aggregate fractions
were then used to re-compose model soils. In order to preserve the original soil structure, the remaining
aggregate size fractions were complemented with an inert matrix replacing the removed aggregate size fraction
of the original soil. Model Soil 1 (LA) was composed of soil macroaggregates mixed in a 1:1 (w/w) ratio with
glass beads of 150–250 µm size serving as inert matrix material replacing the microaggregates of the original
soil. Similarly, Model Soil 2 (SA) was composed of soil microaggregates mixed at the same ratio with fine
quartz gravel of 2000–3200 µm size. To generate an even mixture of original soil aggregates and the respective
inert matrix a Turbula mixer (Willy A. Bachofen AG, Muttenz, Switzerland) was used. The physicochemical
properties of the two soils were determined by analysing three random samples of each model soil. Texture of
the complete model soils was determined using the pipette method (Gee and Bauder, 1986) and pH was
measured potentiometrically in a stirred slurry of 10 g soil in 20 ml of 0.01 M $CaCl_2$ . Additionaly $C_{org}$ and $N_{tot}$
were analysed in both aggregate size fractions without the inert material, using the method described by Walthert
et al. (2010). The two model soils displayed very similar physicochemical properties (Table 1), except for the
C:N ratio that was lower in macroaggregates than in microaggregates. The latter was due to the slightly lower
organic carbon content in concert with slightly higher $N_{tot}$ values in the macroaggregates. The high calcium
carbonate ($CaCO_3$) content of the source material of our model soils ($390 \pm 3$ g $CaCO_3$ $kg^{-1}$; Samaritani et al.,
2011) buffered the systems at an alkaline pH of $8.00 \pm 0.02$ for LA and $7.56 \pm 0.01$ for SA respectively (Table 1),
ensuring that the activity of key N-transforming enzymes was not hampered by too low pH, and that the potential



for simultaneous production and consumption of $N_2O$ in our experiment was fully intact (Blum et al., 2018;
Frame et al., 2017).

### 2.2 Mesocosms

For the mesocosm experiments, transparent polyvinyl chloride (PVC) cylinders with polymethyl methacrylate
(PMMA) couplings were used. A mesocosm comprised a bottom column section, containing the soil material
and a drainage layer as described below, and the upper headspace section with a detachable headspace chamber
(Fig. 1). Each column section was equipped with two suction cups (Rhizon MOM Soil Moisture Samplers,
Rhizosphere Research Products, Netherlands; pore size 0.15 μm) for soil solution sampling. The suction cups
were horizontally inserted at 5 cm and 20 cm below soil surface. For redox potential measurements, two custom-
made Pt electrodes (tip with diameter of 1 mm and contact length of 5 mm) were placed horizontally at a 90°
angle to the suction cups at the same depths, with the sensor tip being located 5 cm from the column wall. A
Ag/AgCl reference electrode (B 2820, SI Analytics, Germany) was installed as shown in Fig. 1. A volumetric
water content (VWC) sensor (EC-5, Decagon, USA) was installed 15 cm below the soil surface. To avoid
undesired waterlogging, each column section contained a 5 cm thick drainage layer composed of quartz sand
with the grain size decreasing with depth from 1 mm to 5.6 mm (Fig. 1). The upper cylinder section was
equipped with three way valves for gas sampling, and an additional vent for pressure compensation.

### 2.3. Experimental setup

The mesocosm experiment had a two factorial design where factor 1 (model soil) had two levels
(macroaggregates or microaggregates) and factor 2 (treatment) had three levels (unamened, litter addition and
plant presence) resulting in six treatments, each replicated six times (Table 2). As basic material, each mesocosm
contained 8.5 kg of either of the two model soils. Unamended model soils were used to investigate exclusively
the effect of aggregate size, abbreviated as LAU (large aggregates unamended) and SAU (small aggregates
unamended), respectively. In order to assess detritusphere effects, two sets of mesocosms were amended with
freshly collected leaves of Basket Willow (*Salix viminalis* L.). Those leaves were cut into small pieces,
autoclaved, and then added to the model soil components (8 g $kg^{-1}$ model soil) during the mixing procedure to
create treatments LAL (large aggregates litter) and SAL (small aggregates litter), respectively. A third set of
mesocosms was planted with cuttings collected from the same basket willow (*Salix viminalis* L.) to evaluate the
effects of root–soil interactions in the respective model soils. For each mesocosm (treatments LAP/large
aggregate plants and SAP/small aggregates plants, respectively) one cutting was inserted 10 cm into the soil,
protruding from the surface about 3 cm.
The addition of leaf litter to the model soils led to an increase of $C_{org}$ and total nitrogen (TN) in LAL relative to
LAU by 41 % and 35 %, respectively, and in SAL relative to SAU by 58 % and 44 % respectively. The bulk
density of the unamended model soil SAU ($1.27 \pm 0.01$ g $cm^{-3}$) was slightly higher than the one of LAU ($1.22 \pm$
$0.01$ g $cm^{-3}$; adj. *P*: $< 0.0001$). Regarding the litter addition treatments, the bulk density of LAL ($1.13 \pm 0.01$ g
$cm^{-3}$) was significantly smaller than the one of LAU (adj. *P*: $< 0.0001$), whereas the bulk density of SAL ($1.27 \pm$
$0.02$ g $cm^{-3}$) did not differ significantly from the one of SAU. The soils in the treatments with plants exhibited a
similar bulk density (LAP: $1.23 \pm 0.02$ g $cm^{-3}$; SAP: $1.24 \pm 0.01$ g $cm^{-3}$) as in the respective unamended
treatments.





The experiments were conducted inside a climate chamber set to constant temperature ($20 \pm 1$ °C) and relative
air humidity ($60 \pm 10$%), with a light/dark cycle of 14/10 h (PAR $116.2 \pm 13.7$ µmol m$^{-2}$ s$^{-1}$). The experimental
period was divided into four consecutive phases: The conditioning phase (phase 1) lasted for 15 weeks and
allowed the model soils to equilibrate and the plants to develop a root system. This was followed by the first
experimental phase of nine days (phase 2), serving as a reference period under steady-state conditions. During
phases 1 and 2, the soils were continuously irrigated with artificial river water (Na$^+$: 0.43 µM; K$^+$: 0.06 µM;
Ca$^{2+}$: 1.72 µM; Mg$^{2+}$: 0.49 µM; Cl$^-$: 4.04 µM; NO$_3^-$: 0.16 µM; HCO$_3^-$: 0.5 µM; SO$_4^{2-}$: 0.11 µM; pH: 7.92) via
suction cups, to maintain a volumetric water content of $35 \pm 5$ %. In phase 3, the mesocosms were flooded by
pumping artificial river water through the drainage vent at the bottom into the cylinder (10 mL min$^{-1}$, using a
peristaltic pump; IPC-N-24, Ismatec, Germany) until the water level was 1 cm above the soil surface. After 48 h
of flooding, the water was allowed to drain and the soil to dry for 18 days without further irrigation (phase 4).

### 205    2.4 Sampling and analyses

During the entire experiment, water content and redox potential were automatically logged every 5 minutes
(EM5b, Decagon, USA and CR1000, Campbell scientific, USA, respectively).
At selected time points during the experiment, soil-emitted gas and soil solution were sampled. For N$_2$O flux
measurements, 20, 40 and 60 minutes after closing the mesocosms, headspace gas samples (20 mL) were
collected using a syringe and transferred to pre-evacuated exetainers. The samples were analyzed for their N$_2$O
concentration using a gas chromatograph (Agilent 6890, Santa Clara, USA; Porapak Q column, Ar/CH$_4$ carrier
gas, micro-ECD detector). Measured headspace N$_2$O concentrations were converted to moles using the ideal gas
law and headspace volume. The N$_2$O efflux rates were calculated as the slope of the linear regression of the N$_2$O
amounts at the three sampling times, relative to the exposed soil surface area (Fig. 1, Shrestha et al., 2012).
For soil water sampling, 20 mL of soil solution were collected using the suction cups. Water samples were
analyzed for dissolved organic carbon (DOC) and TN concentrations with an elemental analyzer (Formacs$^{HT/TN}$,
Skalar, The Netherlands). Nitrate and ammonium concentrations were measured by ion chromatography (IC 940,
Metrohm, Switzerland), and nitrite concentrations were determined photometrically (DR 3900, Hach Lange,
Germany).

### 220    2.5 Data analyses

Differences among the six treatments on individual sampling dates in N$_2$O fluxes, DOC and N-species
concentrations in soil solution were tested for significance using the non-parametric Kruskal–Wallis test
followed by Dunn's post hoc test. To estimate the total amount of N$_2$O emitted during the period of enhanced
N$_2$O fluxes in phase 4, Q$_{tot}$, the N$_2$O fluxes between day 11 and 25 of the experiment were integrated as follows:
$$Q_{tot} = \frac{1}{2} \sum_{n=1}^{n_{max}} [\Delta_n \times (q_n + q_{n+1})], \qquad\qquad (1)$$
where $\Delta_n$ is the time period between the n$^{th}$ and the n+1$^{th}$ measurement, and q$_n$ and q$_{n+1}$ the mean flux on the n$^{th}$
and n+1$^{th}$ measurement day, respectively. "n=1" refers to day 11, and n$_{max}$ to day 25 of phase 4. The integrated
N$_2$O flux data were tested for differences between treatments and model soils by performing a two way ANOVA
and the Tukey's honestly significant difference (HSD) post hoc test. No data transformation was necessary, since
the inspection of residuals of the ANOVA model and the application of the Shapiro–Wilk normality test revealed





that the values follow a Gaussian distribution. Significance and confidence levels were set at $\alpha < 0.05$. For the
statistical analyses we used Graphpad Prism (GraphPad Software Inc., 2017) and R (R Core Team, 2018).

## 3. Results

### 3.1 Soil moisture and redox potential

During phase 1 and 2, saturation levels stabilized at $53.0 \pm 2.1\%$ WFPS (water filled pore space) in the
treatments with LA soils, and were slightly higher in SA treatments ($57.8 \pm 2.0\%$) (Fig. 2). The flooding of the
mesocosms for 48 h with artificial river water raised the WFPS for all LA soils to $87.8 \pm 0.1\%$, significantly
exceeding the increase of WFPS in SA soils ($80.6 \pm 0.1\%$). The water release from the system after the
simulated flood resulted in an immediate drop of the WFPS, except for the LAU treatment (Fig. 2). This was
followed by slow drying for 1 week, and a more marked decrease in WFPS during the second week after the
flood. During the latter period, the plant treatments dried faster than the other treatments. As a result, at the end
of the experiment, WFPS was still above pre-flood values in unamended and litter treatments, while WFPS
levels in the treatments with plants were lower than before the flooding.
The time course of the redox potential measured in 5 cm and 20 cm depth exhibited distinct patterns depending
on the respective model soil (Fig. 3). In all treatments, flooding induced a rapid decrease of the redox potential to
values below 250 mV within 36 hours. Upon water release, the redox potential returned rapidly to pre-flood
values at both measurement depths only in SA soils. In the LA treatments (most pronounced in LAL), soils at 20
cm depth underwent a prolonged phase of continued reduced redox condition, returning to the initial redox levels
only towards the end of the experiment.

### 3.2 Hydrochemistry of soil solutions

Considering individual treatments, DOC concentrations varied only little with time. Yet, the DOC concentrations
were generally much higher in treatments with LA than with SA soils. Nitrate was the most abundant dissolved
reactive N species in the soil solution, with pre-flood concentrations of 1 to 5 mM (Fig. 4d–f). In the unamended
and plant treatments, $NO_3^-$ concentrations were markedly higher in SA than in LA soils, whereas they were
similar in both litter addition treatments. Two distinct temporal patterns in the evolution of $NO_3^-$ concentration
could be discerned. In the unamended and litter-addition treatments, $NO_3^-$ concentrations decreased after the
flooding, consistently reaching a minimum on day 19, in the case of the litter treatments below the detection
limit of 0.2 µM, before increasing again during the later drying phase (Fig 4d,e). In contrast, in the treatments
with plants, $NO_3^-$ concentrations steadily declined from concentrations of 1–2 mM to around 0.5 mM at the end
of the experiment (Fig. 4f). Nitrite was found at significant concentrations only in LA soils, with highest
concentrations in the LAU treatment right after the flooding (33.6 µM) and decreasing concentrations throughout
the remainder of the experiment (Fig. 4g–i). In SA soils $NO_2^-$ concentration was always < 5 µM, without much
variation. Similarly, in most treatments except SAL, ammonium ($NH_4^+$) concentrations were < 10 µM, and
particularly towards the end of the experiment very close to the detection limit (Fig. 4j, 4l). In the SAL treatment,
$NH_4^+$ concentrations peaked 5 d after the flood with concentrations of around 70 µM (Fig. 4k).

### 3.3 Nitrous oxide emissions



During phase 2 (i.e., before the flooding), $N_2O$ fluxes were generally low (< 1 μmol m$^{-2}$ h$^{-1}$; Fig. 2), however,
fluxes in the LAL treatment were significantly higher than in the other treatments (adj. P = 0.002–0.039; Fig. 2).
The flooding triggered the onset of a "hot moment", defined here as period with strongly increased $N_2O$
emissions, which lasted for about one week independent of the treatment (Fig. 2). The maximum efflux was
observed immediately after the flood. The subsequent decline in $N_2O$ emission rates followed different patterns
among the various treatments. Normalizing the $N_2O$ flux to the maximum measured efflux for each replicated
treatment revealed a slower decrease with time for the unamended soils than for the litter and plant treatments
(not shown). The strongest peak emissions were observed in the LAL treatment (91.6 ± 14.0 μmol m$^{-2}$ h$^{-1}$; mean
± SD). Throughout most of the drying phase, the LAU and LAL treatments exhibited higher $N_2O$ emissions than
the corresponding SAU and SAL experiments. In contrast, there was no such difference in the treatments with
plant cuttings, and peak $N_2O$ emissions were overall lower than in the other treatments. The integrated $N_2O$
fluxes during the hot moment (days 10 to 25 of the experiment) were significantly higher for the LAU and LAL
than for all other treatments (Fig. 5). Again, there was a significant aggregate size effect in the unamended (adj.
P = 0.045) and litter-addition treatments (adj. P = 0.008). The integrated $N_2O$ emissions in the two plant
treatments did not differ significantly from each other, but were significantly smaller than in the LAU (adj. P =
0.001), and the LAL (adj. P = 0.005) treatments.
**4. Discussion**
In our experiment, we could confirm the occurrence of periods of enhanced $N_2O$ emissions in the drying phase
shortly after flooding, as expected based on previous research (Baldwin and Mitchell, 2000; Groffman and
Tiedje, 1988; Rabot et al., 2014; Shrestha et al., 2012). We observed that the six treatments had a substantial
effect on the magnitude and temporal pattern of $N_2O$ emissions that could only be captured by observations at
relatively high temporal resolution. The fast occurrence of strong $N_2O$ fluxes over a comparatively short period
in the litter-amended treatment on the one side, and the relatively weak response to the flooding in the plant
treatment on the other, suggests complex interactive mechanisms related to distinct microhabitat effects leading
to characteristic periods of enhanced $N_2O$ emission. Rabot et al. (2014) explained $N_2O$ emission peaks during the
desaturation phase with the release of previously produced and entrapped $N_2O$. Such a mechanism may partly
contribute to high $N_2O$ emissions in our experiment initially, but the continuing depletion of $NO_3^-$ and $NO_2^-$
during the phase of high $N_2O$ emissions indicates that the flooding and drying has strong effects on N
transformations mediated by microorganisms in the soil (e.g., the balance and overall rates of nitrification,
nitrifier–denitrification, and denitrification). Hence, physical controls alone clearly do not explain the observed
timing and extent of hot moments with regard to $N_2O$ emission. In the following sections we will discuss how
the effect of flooding on microbial $N_2O$ production is modulated by differential microhabitat formation (and
hence redox conditions) in the various treatments.
**4.1 Effect of aggregate size on $N_2O$ emissions**
Our results indicate that aggregate size is a major factor in modulating soil $N_2O$ emissions. In the unamended
and litter addition treatments, LA model soils exhibited both higher peak and total $N_2O$ emissions during the hot
moment in the drying phase than SA model soils (Figs. 2 and 5). By contrast, in the presence of a growing
willow, there was no detectable effect of aggregate size on the overall $N_2O$ emission (further discussion below).



The aggregate size effects observed in the unamended and litter treatments can be explained by factors
controlling (i) gas diffusion (e.g. water film distribution, tortuosity of the intra-aggregate pore space) and (ii)
decomposition of encapsulated SOM regulating the extent of $N_2O$ formation (Neira et al., 2015). In order to
isolate the effect of aggregate size (i.e., to minimize the effect of other factors that are likely to influence gas
diffusion), we created model soils of similar soil structure (see Materials and Methods). The results of the
particle size analysis confirmed a nearly identical texture of the two model soils, and at least in the unamended
treatments a similar bulk density was achieved. The effect of soil texture and structure should therefore be
similar for both model soils. The same applies to bulk soil chemical properties of the two aggregate size fractions
such as $C_{org}$ content and pH. Therefore, we assume in the following that the differences in $N_2O$ emissions among
the treatments can mainly be attributed to size-related aggregate properties and their interactions with litter
addition or rhizosphere effects.
During phase 3 with near-saturated conditions, no aggregate size effect was observed. High WFPS seem to have
limited the gas diffusion ($O_2$ and $N_2O$) independent of the aggregate size, limiting soil–atmosphere gas exchange
in both model soils equally (Neira et al., 2015; Thorbjørn et al., 2008). As a consequence of inhibited gas
exchange/soil aeration, a sharp drop in the redox potential was observed in all treatments, indicating a rapid
decline in $O_2$ availability to suboxic/anoxic conditions. Together with an incipient decrease in soil solution $NO_3^-$,
this indicates that $N_2O$ production is primarily driven by denitrification in this phase.
The aggregate size effects on the formation of moments of enhanced $N_2O$ emission became evident during the
subsequent drying period. During the initial drying phase, when a heterogeneous distribution of water films
around soil particles/aggregates develops (Young and Ritz, 2000), the macroaggregates in the LA model soils
appear to foster micro-environmental conditions that are more beneficial to $N_2O$ production. This could be
related to the longer diffusive distances for re-entering $O_2$ caused by the higher tortuosity of the intra-aggregate
pore space of macroaggregates, as reported by Ebrahimi and Or (2016). This may have helped to maintain, or
even extend, reducing conditions due to microbial activity inside the core of macroaggregates during drying.
Thus, on the one hand, large aggregates favor the emergence of anoxic microhabitats expanding the zones where
denitrification occurs. On the other hand, the overall higher porosity of the LA soils supports a better aeration in
drained parts of the soil (Sey et al., 2008), and aerobic processes (e.g., nitrification) are supported. As a result,
ideal conditions for spatially coupled nitrification–denitrification are created (Baldwin and Mitchell, 2000;
Koschorreck and Darwich, 1998). Indeed, the emergence of heterogeneously distributed, spatially confined
oxygen minimum zones during soil drying may be reflected by the high variability of the redox conditions
observed in replicate mesocosms and, on average, the tendency towards lower redox potentials for a prolonged
period of time in the subsoils of the LA model soils (Fig. 3 d–f). In this context, the relevance of water films for
the emergence of periods of enhanced $N_2O$ emissions is further highlighted by the fact that elevated flux rates
were only observed as long as the WFPS was above 65%. This is consistent with work by Rabot et al. (2014)
and Balaine et al. (2013), who found similar soil water saturation thresholds for elevated $N_2O$ emissions from
soils, attributing this phenomenon to suboptimal environmental conditions for both nitrification and
denitrification at lower saturation levels.
Given the arguments above, we assume that $N_2O$ emissions during the drying phase originate to a large degree
from heterotrophic denitrification, and that they are governed mainly by the aggregate size dependent redox
conditions within the semi-saturated soils. This conclusion stands in good agreement with findings from Drury et
al. (2004) who found higher production of $N_2O$ due to enhanced denitrification with increasing size of intact




arable soil aggregates in a laboratory incubation study. In contrast, the much lower emissions from the SA
treatments can best be explained by a rapid return to pre-flood, i.e. oxic, conditions in most of the pore space,
under which $N_2O$ production driven by denitrification is inhibited. According to Manucharova et al. (2001) and
Renault and Stengel (1994), aggregates smaller than 200 µm are simply not large (and reactive) enough (i.e.,
molecular diffusive distances for oxygen are too short) to develop suboxic or anoxic conditions in the center, let
alone denitrifying zones. Hence, only a relatively small fraction of the total number of microaggregates in the SA
soils would have been large enough (between 200 and 250 µm) to host denitrification and act as site of anaerobic
$N_2O$ production.

**4.2 Litter effect on $N_2O$ emissions**

We expected that litter addition would increase $N_2O$ emissions from model soils with both small and large
aggregates, as was found earlier (e.g. Loecke and Robertson, 2009; Parkin, 1987). The addition of litter to the
model soils changed the temporal dynamics of the $N_2O$ emission substantially, but its effect on the net integrated
$N_2O$ emission was rather minor (Fig. 5). More precisely, highest peak emission rates of all treatments were
observed in the LAL treatment, but peak emission rates were followed by a faster return to low pre-flood
emission rates in the LAL and the SAL treatments relative to the unamended treatments (Fig 2). This confirms
that surplus organic carbon can, on short-term, boost $N_2O$ emissions, particularly in the large-aggregate
treatment. The fast mid-term return to low $N_2O$ emission suggests that $N_2O$ production by heterotrophic
denitrification either becomes limited by substrates other than carbon, and/or that the carbon added to the soils
affects the redox-biogeochemistry in a way that shifts the balance between $N_2O$ production and consumption in
favor of consumption. Loecke and Robertson (2009) reported similar temporal $N_2O$ emission patterns in field
experiments with litter-amended soil, and attributed the observed dynamic of a rapid decline after peak emission
to an increased demand for terminal electron acceptors during denitrification shortly after the carbon addition.
Nitrate/nitrite limitation leads, under stable anoxic conditions, ultimately to the complete reduction of produced
$N_2O$ to $N_2$ decreasing net $N_2O$ emission. Indeed, the rapid decrease in $N_2O$ emissions after the emission rate peak
in the litter addition treatments was accompanied by the complete depletion of $NO_3^-$ in the soil solution at low
redox potential, suggesting nitrate limitation. The increased demand for electron acceptors can be attributed to
the increased availability of labile C compounds and nutrients provided by the mineralization of litter, and the
concomitant stimulation of aggregate-associated microbial communities during the flooding (Li et al., 2016). At
the same time, the litter-stimulated soil respiration increases the soil's oxygen demand, maintaining stable low
redox conditions for a longer period of time during the drying phase. Since high activity of $N_2O$ reductase
requires very low $O_2$ concentrations (Morley et al., 2008), such conditions may be particularly favorable for
complete denitrification to $N_2$, an additional, or alternative, explanation for the low $N_2O$ emission rates shortly
after the $N_2O$ emission peak.

**4.3 Effects of _Salix viminalis_**

Planted willow cuttings resulted in relatively low maximum $N_2O$ emission rates (LAP: $19.75 \pm 9.31$ µmol $m^{-2}$ $h^{-1}$
; SAP: $15.07 \pm 12.07$ µmol $m^{-2}$ $h^{-1}$; mean ± SD), independent of aggregate size. The high values for WFPS
throughout the hot moment, and a low redox potential in the subsoil, imply optimal conditions for denitrification
or nitrifier denitrification, but compared to unamended and litter-addition treatments, only little $N_2O$ was emitted



(both during peak N$_2$O emission rates and with regards to the integrated N$_2$O flux). *S. viminalis* suppressed peak
N$_2$O emissions, overriding the positive effect of large aggregates on N$_2$O emissions observed otherwise. The
specific mechanisms involved are uncertain. Fender et al. (2013) found in laboratory experiments with soil from
a temperate broad-leaved forest planted with ash saplings N$_2$O fluxes and plant effects very similar to the ones
observed in our study. They attributed reduced N$_2$O emissions in presence of ash partly to plant uptake of
nutrients that reduced NO$_3^-$ availability to denitrifiers. Fast-growing plant species like *Salix* are particularly
effective in removing soil inorganic N (Kowalik and Randerson, 1994). Such a causal link between reduced N$_2$O
emissions and plant growth is, however, not supported by our data. More precisely, the NO$_3^-$ concentrations
during the hot moment of N$_2$O emissions were always relatively high (> 0.5 mM) and above the levels observed
in the litter treatments.
An alternative explanation for the reduced N$_2$O emissions in the plant treatments could be rhizosphere aeration
by aerenchyma, a physiological trait of *Salix viminalis* roots, which prevents the formation of anoxia in their
close vicinity (Blom et al., 1990; Randerson et al., 2011). Thus, while aerenchyma in general aid in the gas
exchange between the soil and the atmosphere, and would per se accelerate transport of N$_2$O from soils to the
atmosphere, they also inhibit anaerobic N$_2$O production by aerating the rhizosphere. Indeed, redox potentials in
the topsoil were higher in SAP and LAP compared to the other treatments. By contrast, the redox potential in the
saturated subsoil below was even lower than observed for the unamended soils. This indicates that the aeration
effect by aerenchyma is constrained to the upper soil, or is, in the deeper soil portions, compensated by
respiratory rhizosphere processes. According to Fender et al. (2013), in vegetated soils, microbial respiration is
stimulated by deposition of root exudates, which in concert with root respiration in a highly saturated pore space,
leads to severe and ongoing oxygen depletion. Again, N$_2$ and not N$_2$O is the dominant final product of
denitrification,under the stable anoxic condition produced this way, and N$_2$O emissions will be low.
**5. Conclusions**
In this study, we investigated the distinct effects of aggregate size, surplus organic carbon from litter and
vegetation on N$_2$O emission from model soils after flooding. .Flooding and drying were always associated with
hot moments of N$_2$O production, most likely due to heterotrophic denitrification as result of suboxic O$_2$ levels at
high WFPS. Our results demonstrate that aggregate size is a very important factor in modulating N$_2$O emission
from soils under changing pore space water saturation. Aggregates of a diameter > 250 μm appear to foster
suboxic microhabitats that favor denitrification and associated N$_2$O emission. This soil aggregate size effect may
be amplified in the presence of excess carbon substrate, as long as heterotrophic denitrification as the main N$_2$O
producing process is not electron acceptor limited, and extremely reducing conditions in organic rich soils do not
promote complete denitrification leading to a further reduction of N$_2$O to N$_2$. On the other hand, the higher
porosity of the soils with macroaggregates may aid in the formation of microsites at the surface of aggregates
where nitrification is re-initialized during drying, supporting favorable conditions for spatially coupled
nitrification–denitrification. The mechanisms by which processes in the rhizosphere of *Salix viminalis* effectively
suppress N$_2$O emissions, and thus mask any aggregate size effect, remain ambiguous. Distinct physiological
features of *Salix viminalis*, its root metabolism, in combination with microbial respiration can lead to the
simultaneous aeration of some parts of the rhizosphere, and the formation of strongly reducing zones in others.
In both cases, redox conditions seem to be impedimental for extensive net N$_2$O production.





Our results demonstrate the importance and complexity of the interplay between soil aggregate size, labile
organic C availability, respiratory processes in the rhizosphere, and plant-induced aeration of soils under
changing soil water content. Those interactions emerged as modulators of $N_2O$ emissions by controlling the $O_2$
distribution in the soil matrix. Indeed, $O_2$ appears as the unifying master variable that ultimately sets the
boundary conditions for $N_2O$ production and/or consumption.
The main scope of this work was to expand our knowledge on the controls on net $N_2O$ emissions from floodplain
soils. The systematic relationships observed in this study are likely to help anticipating where and when hotspots
and hot moments of $N_2O$ emissions are most likely to occur in hydrologically dynamic soil systems like
floodplain soils. Further understanding of the complex interaction between plants and soil microorganisms, the
detritusphere, and soil aggregation, as well as their influence on N turnover and $N_2O$ accumulation in soils,
should focus on how the parameters tested affect the actual activity of the nitrifying and denitrifying
communities, with an in-depth investigation into the biogeochemical pathways involved.
*Data availability.* Data will be openly available at https://datadryad.org/
*Competing interests.* The authors declare that they have no conflict of interest.
*Authors contributions.* The initial concept of the experiment was developed by JL, MFL and PAN. ML planned
the experiment in detail, set it up and performed it. PAN supervised the measurement of $N_2O$ gas concentrations,
whereas ML conducted all other measurements and data analyses. ML wrote the manuscript with major
contributions by JL, MFL and PAN.
*Acknowledgements.* The authors thank the Department of Evolutionary Biology and Environmental Studies of
the University of Zurich and René Husi for performing the GC measurements. We are also very grateful to the
Environmental Geoscience research group in the Department of Environmental Sciences of the University of
Basel and Judith Kobler–Waldis for helping us with the IC measurements. We thank the Central Laboratory and
Daniel Christen, Roger Köchli and Noureddine Hajjar of the Swiss Federal Institute for Forest, Snow and
Landscape Research (WSL) for assistance with chemical analyses. This study was funded by the Swiss National
Science Foundation (SNSF) under the grant number 200021_147002 as well as by financial resources of WSL
and the University of Basel.

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





**Table 1: Physicochemical properties of the two aggregate size fractions (macroaggregates and microaggregates) and**
**added leaf litter. $C_{org}$ and $N_{tot}$ of the aggregates was measured in triplicates. The leaf litter was analyzed in four**
**replicates. Final pH and texture of model soil 1 and 2 was measured in duplicates (means ± SD)**

|  |  | Macroaggregates | Microaggregates | Litter (*Salix v.* L.) |
|---|---|---|---|---|
| $C_{org}$ | g kg$^{-1}$ | 19.22  ± 0.55 | 21.56  ± 2.39 | 459.9  ± 2.55 |
| Total N | g kg$^{-1}$ | 1.58  ± 0.02 | 1.35  ± 0.14 | 27.39  ± 0.15 |
| C:N ratio |  | 12.16  ± 0.22 | 15.99  ± 0.71 | 16.79  ± 0.06 |
|  |  | Model soil 1 | Model soil 2 |  |
| pH (CaCl$_2$) |  | 8  ± 0.02 | 7.56  ± 0.01 |  |
| sand | % | 71.25  ± 0.05 | 70.7  ± 0.50 |  |
| silt | % | 20  ± 0.30 | 21.1  ± 0.60 |  |
| clay | % | 8.75  ± 0.25 | 8.2  ± 0.10 |  |


**Table 2: Overview of treatments in the flooding–drying experiment**

|  | LAU | SAU | LAL | SAL | LAP | SAP |
|---|---|---|---|---|---|---|
| Model Soil 1 (LA) | + | - | + | - | + | - |
| Model Soil 2 (SA) | - | + | - | + | - | + |
| Leaf litter (*Salix v.*) | - | - | + | + | - | - |
| *Salix v.* | - | - | - | - | + | + |




## Figure Captions

**Figure 1:** Schematic of a mesocosm with gas sampling valves (1), Ag/AgCl reference electrode (2), Pt redox electrodes (3), suction cups (4), volumetric water content sensors (5), vent (6), and water inlet/outlet (7). The top part is only attached during gas sampling.

**Figure 2:** Mean $N_2O$ emission during the flooding–drying experiment from large-aggregate model soil (LA; filled circles) and small-aggregate model soil (SA, open circles), and corresponding water-filled pore space (WFPS) in LA (filled triangles) and SA (open triangles). Unamended soils (A), litter addition (B) and plant treatment (C). Flooding phase indicated by the grey area. Symbols indicate means; error bars are SE; n= 6.

**Figure 3:** Redox potential relative to standard hydrogen electrode during the flooding–drying experiment in 5 cm and 20 cm depth (mean ± SE; n=6). Unamended soils (a and d, respectively), litter addition (b and e, respectively), plant treatment (c and f, respectively). LA (filled circles) and SA (open circles); the dotted line at 250 mV marks the threshold, below which denitrification is expected to occur.

**Figure 4:** DOC (circles), nitrate (squares), nitrite (diamonds) and ammonium (triangles) concentrations in pore water during the flooding–drying experiment. LA (filled symbols) and SA (empty symbols). Unamended soils (a, d, g and j, respectively), litter addition (b, e, h and k, respectively) and plant treatment (c, f, j and l, respectively).; (mean ± SE; n=6).

**Figure 5:** Integrated $N_2O$ fluxes over the 14 days period of elevated $N_2O$ emissions in the drying phase of the flooding– drying experiment (mean ± SE; n= 6). Black bars represent model soil 1 (macroaggregates 250-4000μm) whereas model soil 2 (microaggregates < 250μm) is depicted as white bars. Significant differences among the six treatments are denoted by different lower case letters at adj. P < 0.05.



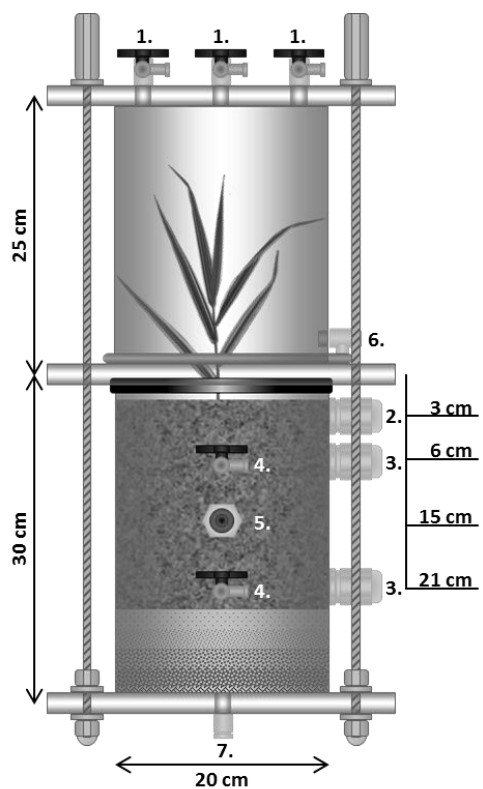


Figure 1





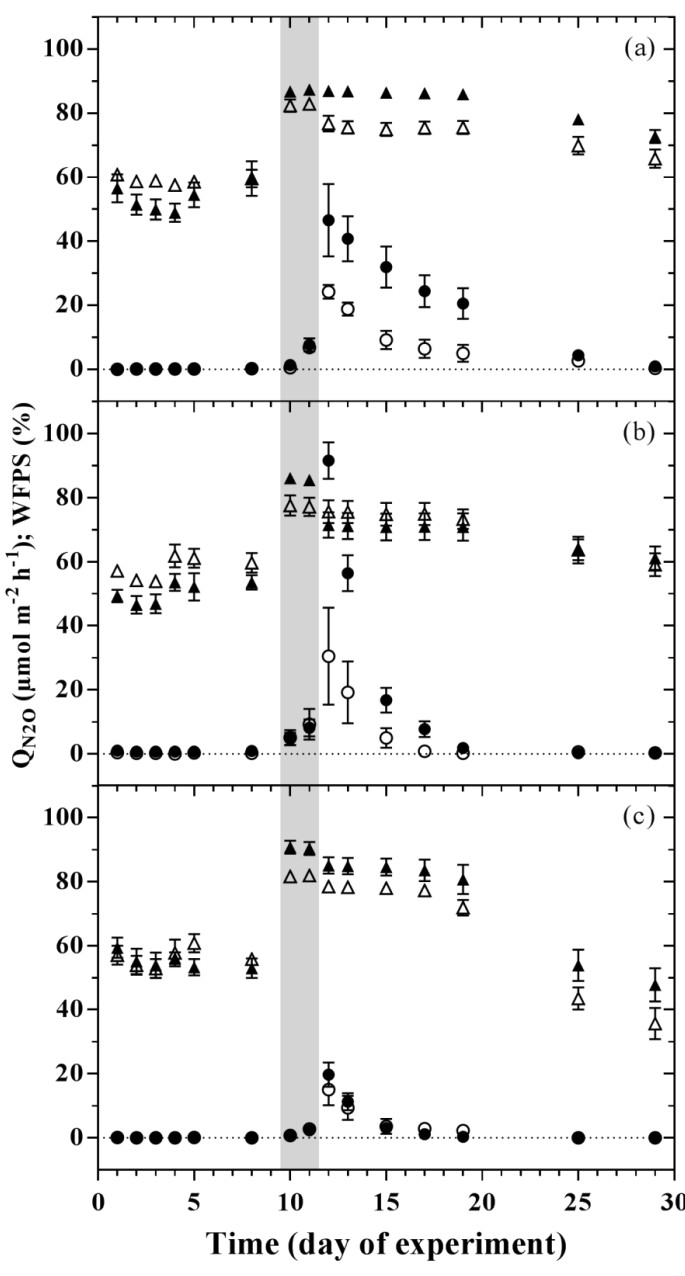


Figure 2



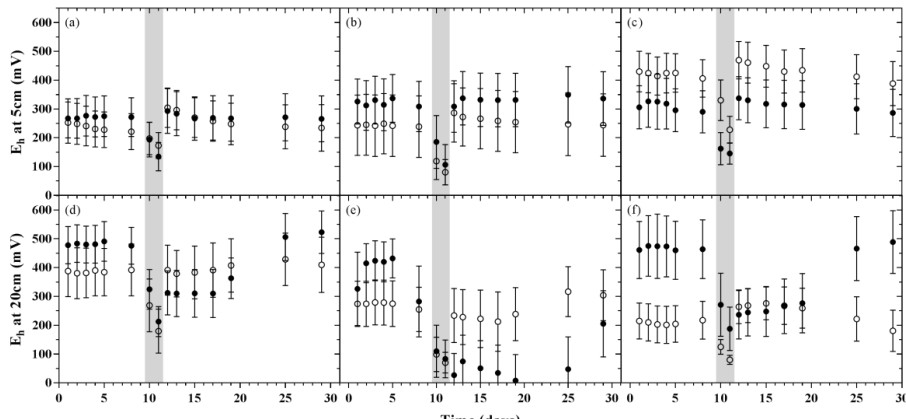


Figure 3



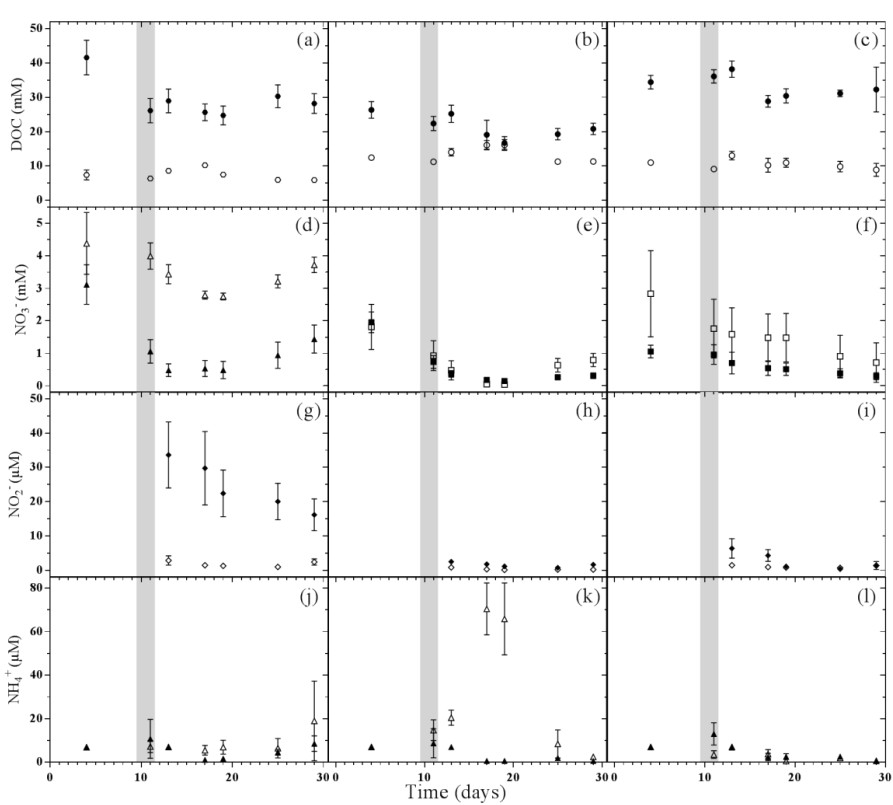


Figure 4





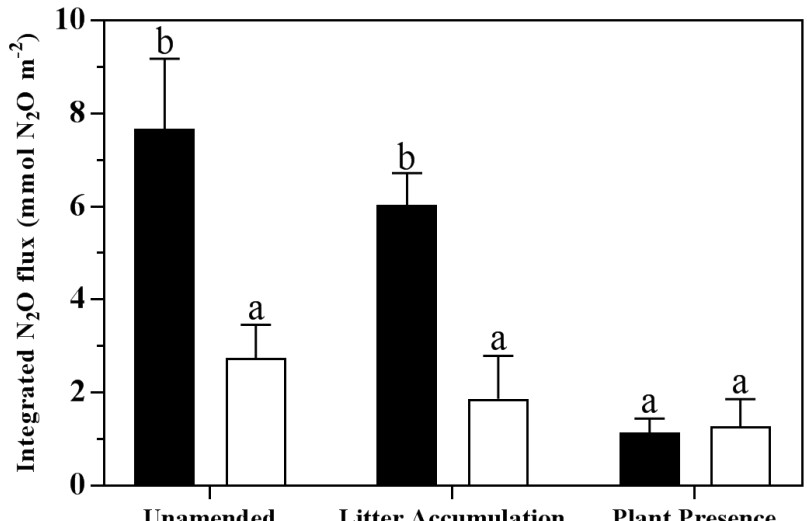


Figure 5