# Peer review of "Alteration of nitrous oxide emissions from floodplain soils by 1"

_Biogeosciences, 2018_

## Referee Comment (RC1) · Anonymous Referee #1 · 17 Jul 2018

This work studied the effects of drying-wetting, soil aggregate size, litter addition and plant on N2O flux from floodplain soils. The authors used model soils and mesocosm experiments to conduct the research. As far as I can say, there are still much more space can be improved for this manuscript.

In general, it is interesting to know how soil N2O flux are controlled by different environmental factors. However, there are already many studies conducted in no matter drying-wetting and soil aggregation, or litter addition and vegetation effects. What the knowledge gaps do you want to fill? It should be clarified in the introduction part.

Here are technical questions:

1. Line 14-15, it is not accurate to write the buried organic matter and rhizosphere processes. Actually, the experiments were about litter addition and plant vegetation. It still takes several steps from litter to organic matter. And also, you didn't took the rhizosphere samples.

2. L148, for soil pH measurement, normally it is 10 g soil was mixed with 25 mL solution. The authors used 20 mL of solution, any references? The solution can be water or CaCl2, as far as I know, for alkaline soil, it it better to use water. In this study, the soil pH were ~ 8, any reasons to choose CaCl2?

3. Have the authors ever considered the emission/uptake of N2O by the aboveground of plant? There are already many studies in this field, such as: Smart D R, Bloom A J. Wheat leaves emit nitrous oxide during nitrate assimilation[J]. Proceedings of the National Academy of Sciences, 2001, 98(14): 7875-7878. In this study, the authors measured N2O flux from the mesocosm have both soil and plant. This flux cannot be called soil flux, but may be soil/plant flux?

4. L274, the author can show the data in support information.

5. L313-315, the authors didn't check the statistics difference of soil chemical/physical properties between different treatments. Therefore, the hypothesis is not really correct before statistics analysis were done.

6. L346-347, Actually WFPS-SA value were not decreased to pre-flood even until the end of experiments (Fig. 2 a and b). The explanation might be low diffusion rate of N2O in SA treatments caused reduction of N2O to N2?

7. L409, delete one dot

8. L457, delete DOI

9. Table 2, it would be better to explain the meanings of LAU, SAU….in the table caption.

10. L638-639, no dotted line in Fig. 3?

11. Fig. 2, it would be better to put WFPS in the right Y axis. And put WFPS-LA, WFPS-SA….in the figure legend.

12. Fig. 3e, the data are not completely shown.

13. Fig. 4, would be better to have the same unit (μM) for nitrate and nitrite/ammonium.

---

## Referee Comment (RC2) · Y. A. Teh (Referee) · 25 Jul 2018

GENERAL COMMENTS

This is a creative and interesting process-based experiment that uses different aggregate treatments (i.e. micro- versus macro-aggregate dominated) and plant-soil treatments (i.e. a gradient of "plant influence," from rhizosphere to detritus-affected soil to plant-free soil) to determine how differences in soil structure and various levels of plant influence potentially influence N2O dynamics in soil. The factorial experimental design is powerful because it enables the investigators to assess not only main effects, but also evaluate the potential importance of synergistic effects among different treatments. Overall, it is my view that this paper was clearly written, with a well-justified experimental design, and a logical analysis of the data. The introduction to the paper clearly explains the basis and wider significance of this research, while the methods section explains the overall approach taken with clarity. The results section documents the main findings of the work succinctly, while the discussion takes a reasonable (and not overly speculative) approach to data interpretation, informed by the authors' grasp of the current literature. The investigators' comprehensive measurement of a range of environmental parameters is to be commended and enables them to make logical inferences about the role of different treatments and environmental factors in regulating N2O dynamics during different parts of the simulated water cycle. In particular, the investigators make good use of redox potential measurements to evaluate how changes in redox/O2 availability could be driving N dynamics along the "plant influence" gradient that they have created in the laboratory.

However, while I am generally supportive of this research and believe it will make a valuable contribution to the wider body of knowledge on this topic, I do have a few general remarks that I believe need to be addressed before this paper can go forward to publication. First, I think the authors need to be open and transparent about the potential limitations of their research. For example, the soil structure treatments represent two extremes (large versus small aggregates), whereas in reality micro- and macro-aggregates would be mixed together. The authors need to explain how their experimental treatment could relate or correspond to real-world conditions, drawing if possible on pre-existing field or laboratory data (see points 1 and 5 below). Likewise, the authors need to be clearer about the limitations underlying their rhizosphere (Salix) treatment. It is difficult to generalise more widely about the effects of plant rhizospheres on N dynamics without examining a range of different plants (including single and multi-species mixtures), in order to tease-apart individual species effects from generic rhizosphere effects (see point 6 below); I think it is important, in the revised version of this text, that the authors acknowledge this limitation and spend a bit more time exploring what they believe could be more widely generalisable from their

study, rather than what is species-specific.

Second, I do not believe that the authors have fully exploited their experimental design in the analysis of their data, and sincerely believe that more could be done to examine these data in greater depth. For example, as mentioned above, one of the strengths of a factorial experimental design is that the investigators can establish if there are synergistic interactions among different experimental treatments (e.g. aggregate X rhizosphere effects). However, the investigators do not appear to have examined if interactions among treatments occurred, or at least these findings are not reported if these tests were conducted. Moreover, I would suggest that the authors try more complex multivariate models to analyse their data; for instance, using approaches such as analysis of co-variance (ANCOVA), generalized linear models, or mixed effects models. The benefit of these more comprehensive multivariate models is that they enable the investigator to establish the relative importance of different treatments and continuous environmental variables in regulating flux.

Third, I agree with the first referee that the authors need to spend a bit more time clearly highlighting what knowledge gaps this paper fills. As the first referee indicates, there are already existing studies that have examined the individual effects of all the variables discussed here. In order to make this paper more impactful, the authors need to articulate how this specific study is unique or advances our current state-of-knowledge (e.g. does the factorial design add knowledge or insight?).

Specific comments are provided in the section below.

SPECIFIC COMMENTS

1. Lines 136-137: For experimental purposes, the investigators have created quasi-artificial system conditions, with treatments either containing macro- or micro-aggregates. While I fully understand why this was done, it would be useful to understand (even qualitatively) how close or far from reality these treatments are. For example, what was the proportion of macro- and micro-aggregates under natural conditions?

2. Line 173: Clarity of expression; consider revising this section to read "The mesocosm experiment had a factorial experimental design consisting of two factors (model soil and plant-soil treatment), with the first factor containing two levels (macroaggregates, microaggregates) and the second factor containing three levels (unamended, litter added, plant present). This experimental design resulted in six treatments, each replicated six times."

3. Line 179-180: What was the rationale for autoclaving the leaves? Under natural conditions, these leaves would contain their own microbial community which could contribute to N2O dynamics, and autoclaving means that the results will be biased towards the activity of the soil community (or, spore-forming phyllosphere microbes able to resist the effects of autoclaving).

4. Lines 221-232: Further detail on the statistical analyses are required here. For example, what were the independent variables used in the ANOVA? Did the model include interaction terms? Given that sampling was conducted over different periods of time, did the authors use a repeated measures ANOVA, to account for the effects of time?

5. Lines 300-353: This is an interesting and well-written part of the discussion. However, I do think that this part of the discussion could be improved by trying to link back the findings from the experiment to natural conditions (see point 1). For example, under natural conditions, what is the relative distribution of macro- or micro-aggregates? Based on your understanding/knowledge of the natural aggregate distributions, what patterns or processes do you think will dominate in a natural setting? While I realise this might be somewhat speculative (unless other data, such as field measurements, are available), I think it's an important talking point, as it will enable the reader to relate these findings (derived under somewhat artificial conditions) to the real world.

6. Lines 380-406: The discussion of potential direct and indirect effects facilitated by

the presence of an active root system is interesting and well-reasoned. However, I was left wondering as to how generalizable these findings are, given the wide range of traits displayed by different plants? I.e. to what extent are the trends identified here unique to Salix, and to what extent are these patterns more widely generalizable? I think it is important that the authors develop this section a bit further, in particular acknowledging this limitation more frankly.

Yit Arn Teh, School of Biological Sciences, University of Aberdeen, Aberdeen AB24 3UU, Scotland, UK

---

## Author Response (AR1)

Response to comments by anonymous referee #1

We thank the reviewer for her/his insightful comments and questions.

This work studied the effects of drying-wetting, soil aggregate size, litter addition and plant on $N_2O$ flux from floodplain soils. The authors used model soils and mesocosm experiments to conduct the research. As far as I can say, there are still much more space can be improved for this manuscript.

In general, it is interesting to know how soil $N_2O$ flux are controlled by different environmental factors. However, there are already many studies conducted in no matter drying-wetting and soil aggregation, or litter addition and vegetation effects. What the knowledge gaps do you want to fill? It should be clarified in the introduction part.

Reply: We concur with the reviewer that the specific objectives of this study were not sufficiently well stated. While the effects of microhabitats related to soil aggregates, the detritusphere and plant-soil interactions in the rhizosphere on $N_2O$ emissions from soils have been studied individually, little is known about their relative effects and interactions. In a mesocosm study, we investigated this aspect for the hot moments of $N_2O$ emissions from floodplain soils during the drying phase after flooding. In particular, aggregate size effects have not been investigated in this context (as stated on lines 80ff). A particular novel aspect of the study is the minimization of the potentially confounding factor "soil structure" by mixing a given aggregate size fraction with inert material replacing the removed smaller or larger fraction. As stated on line 71ff, previous studies employing isolated aggregate size fractions have provided partially inconsistent results.
The innovative aspects of objectives,will be further clarified in the introduction of the revised manuscript, with emphasis on the relevance of the research, and addressing also the potential with regards to filling knowledge gaps.

Here are technical questions:

1. Line 14-15, it is not accurate to write the buried organic matter and rhizosphere processes. Actually, the experiments were about litter addition and plant vegetation. It still takes several steps from litter to organic matter. And also, you didn't took the rhizosphere samples.

R1: We agree that the term "buried organic matter" is too unspecific. Although, sensu stricto, litter is "organic matter" as well, it indeed might be confused with further decomposed and transformed "soil organic matter". We therefore have replaced "buried organic matter" with "buried litter". We checked the entire manuscript and this is the only place where we used this rather unspecific term.
We also agree that at this point "rhizosphere processes" should be replaced by "plant-soil interactions", even though in the later discussion mainly rhizosphere processes per se are invoked to explain the observed plant effects.

2. L148, for soil pH measurement, normally it is 10 g soil was mixed with 25 mL solution. The authors used 20 mL of solution,any references? The solution can be water or CaCl2, as far as I know, for alkaline soil, it it better to use water. In this study, the soil pH were ~ 8, any reasons to choose CaCl2?

R2: There are several soil-to-solution ratios recommended in the literature, among them also 1:2.5 (Blume et al. 2010. Scheffer/Schachtschabel – Lehrbuch der Bodenkunde, 16[th] ed., p. 151), or 1:1 (Thomas G.W. 1996. «Soil pH and Soil acidity» In: Sparks et al. (eds.) Methods of soil analysis – 3. Chemical methods. SSSA Book Series 5, pp. 475ff.). A soil-to-solution ratio of 1:2 for mineral soil samples – as has been used in our laboratories since more than 30 years – is also recommended by one of the newest method handbooks: Hendershot et al.

(2008) "Soil reaction and exchangeable acidity" In: Carter, M.R. (ed.) Soil sampling and methods of analysis. 2$^{nd}$ ed., Can. Soc. Soil Sci., chapter 16. Furthermore, this handbook, citing several individual studies, recommends $CaCl_2$ as suspending solution with several advantages over water, in particular also for agricultural soils whose pH is often comparatively high. There is no mentioning in this, or any of the other cited references, of a disadvantage in using $CaCl_2$ for carbonate containing soils.

More generally, soils are heavily buffered systems and the measured pH should be virtually independent of such small variation in ionic strength.

3. Have the authors ever considered the emission/uptake of N2O by the aboveground of plant? There are already many studies in this field, such as: Smart D R, Bloom A J. Wheat leaves emit nitrous oxide during nitrate assimilation[J]. Proceedings of the National Academy of Sciences, 2001, 98(14): 7875-7878. In this study, the authors measured N2O flux from the mesocosm have both soil and plant. This flux cannot be called soil flux, but may be soil/plant flux?

R3: In the introduction (line 96ff) we considered potential bypassing of the soil matrix by $N_2O$ fluxes via plant-internal aeration channels (aerenchyma). This phenomenon is well documented for *Poaceae* such as the Genus *Oryza* or *Phalaris arundinacea*. However, for willows (*Salix sp.*) such a process has, to our knowledge, not been documented yet. Although, considering that also adventitious roots of *Salix* species contain aerenchyma, we cannot exclude this process to occur in our case, our results do not indicate an enhanced $N_2O$ emission via the plant, since we observed the lowest flux rates as well as lowest total integrated emissions in the mesocosms with plants. Therefore we conclude that in our experiment, such a process, if present, was of minor importance in terms of modulating net $N_2O$ fluxes to the atmosphere. However, we agree that the possibility that part of the $N_2O$ fluxes from the planted soils occurred via plant-internal channels should be mentioned in discussion section 4.3. We also agree that emission fluxes should be termed "soil/plant flux" or "ecosystem flux" instead of "soil flux". Although nowhere in the manuscript we have used the term "soil flux", we agree that we need to clarify at respective prominent places in the manuscript that in the case of the treatments with willow emissions/fluxes relate to the whole soil/plant system and not to the soil alone.

4. L274, the author can show the data in support information.

R4: we will upload a file containing the supplementary information and adjust the text accordingly.

5. L313-315, the authors didn't check the statistics difference of soil chemical/physical properties between different treatments. Therefore, the hypothesis is not really correct before statistics analysis were done.

R5: The comparison of the initial physicochemical properties by t-tests with Welch's correction showed statistically significant differences for the C:N ratio and pH. However, C:N ratios of 12 and 16 can be considered ecologically similar in terms of soil organic matter degradability, in particular since both Corg and total N do not differ that much. The higher pH in the macroaggregated model soil is probably due to a higher carbonate content, which also is not expected to strongly affect biogeochemical processes of the N cycle. These remarks will be added in the revised manuscript, and a new column will be added in table 1 with the results of the statistical analyses.

6. L346-347, Actually WFPS-SA value were not decreased to pre-flood even until the end of experiments (Fig. 2 a and b). The explanation might be low diffusion rate of N2O in SA treatments caused reduction of N2O to N2?

R6: Considering the high WFPS in the SAU treatment, the referee's remark represents a valid explanation for the observed low fluxes under the given circumstances. However, the relatively high redox potentials, which we invoke here, argue against sufficient anoxia for complete reduction of $N_2O$ to $N_2$. Nevertheless, we will include this aspect in the discussion in section 4.1. of the revised manuscript.

7. L409, delete one dot

8. L457, delete DOI

9. Table 2, it would be better to explain the meanings of LAU, SAU....in the table caption.

10. L638-639, no dotted line in Fig. 3?

11. Fig. 2, it would be better to put WFPS in the right Y axis. And put WFPS-LA, WFPS-SA....in the figure legend.

12. Fig. 3e, the data are not completely shown.

13. Fig. 4, would be better to have the same unit (µM) for nitrate and nitrite/ammonium

R7-13: the authors consent with all these remarks and will make changes to the revised manuscript accordingly.

Response to comments by Y. A. Teh (Referee)

We thank the reviewer for his supportive evaluation, insightful comments and questions. Addressing them will strongly improve the manuscript.

GENERAL COMMENTS
This is a creative and interesting process-based experiment that uses different aggregate treatments (i.e. micro- versus macro-aggregate dominated) and plant-soil treatments (i.e. a gradient of "plant influence," from rhizosphere to detritus-affected soil to plant-free soil) to determine how differences in soil structure and various levels of plant influence potentially influence N2O dynamics in soil. The factorial experimental design is powerful because it enables the investigators to assess not only main effects, but also evaluate the potential importance of synergistic effects among different treatments. Overall, it is my view that this paper was clearly written, with a well-justified experimental design, and a logical analysis of the data. The introduction to the paper clearly explains the basis and wider significance of this research, while the methods section explains the overall approach taken with clarity. The results section documents the main findings of the work succinctly, while the discussion takes a reasonable (and not overly speculative) approach to data interpretation, informed by the authors' grasp of the current literature. The investigators' comprehensive measurement of a range of environmental parameters is to be commended and enables them to make logical inferences about the role of different treatments and environmental factors in regulating N2O dynamics during different parts of the simulated water cycle. In particular, the investigators make good use of redox potential measurements to evaluate how changes in redox/O2 availability could be driving N dynamics along the "plant influence" gradient that they have created in the laboratory.

However, while I am generally supportive of this research and believe it will make a valuable contribution to the wider body of knowledge on this topic, I do have a few general remarks that I believe need to be addressed before this paper can go forward to publication. First, I think the authors need to be open and transparent about the potential limitations of their research. For example, the soil structure treatments represent two extremes (large versus small aggregates), whereas in reality micro- and macro-aggregates would be mixed together. The authors need to explain how their experimental treatment could relate or correspond to real-world conditions, drawing if possible on pre-existing field or laboratory data (see points 1 and 5 below).

R I: We agree that including a discussion of the implicit limitations of our experimental approach with respect to natural conditions will contribute to a better evaluation of the results of our study, and we thus will include this in a revised version of the manuscript.
By investigating two pedogenetically well-defined aggregate size fractions (4000 – 250 μm and 250 – 0 μm; Tisdall and Oades, 1982) separately – but with soil structure kept similar by replacing the removed fraction by inert material of the same size - , we aimed at evaluating the individual potential of these fractions to offer conditions for the soil microbial community to form $N_2O$. Following the reviewer's suggestion, we propose to include a discussion of how these conditions relate to real-world conditions as follows. As detailed in our response R1 below, these two size fractions represent significant "components" both of our investigated original soil and of most other soils. However, we intentionally excluded interactions between the two soil aggregate size factions to assess the individual potential of each faction separately. Therefore we can neither assess any interactions between large and small aggregates, nor such with soil structures larger than 4mm, which all may also be important for $N_2O$ emissions under natural conditions. Since we have no data related to this, we prefer not to speculate about such effects in our paper.

Likewise, the authors need to be clearer about the limitations underlying their rhizosphere (Salix) treatment. It is difficult to generalise more widely about the effects of plant rhizospheres on N dynamics without examining a range of different plants (including single and multi-species mixtures), in order to tease-apart individual species effects from generic rhizosphere effects (see point 6 below); I think it is important, in the revised version of this text, that the authors acknowledge this limitation and spend a bit more time exploring what they believe could be more widely generalisable from their study, rather than what is species-specific.

R II: For a reply the reader is kindly referred to R6

Second, I do not believe that the authors have fully exploited their experimental design in the analysis of their data, and sincerely believe that more could be done to examine these data in greater depth. For example, as mentioned above, one of the strengths of a factorial experimental design is that the investigators can establish if there are synergistic interactions among different experimental treatments (e.g. aggregate X rhizosphere effects). However, the investigators do not appear to have examined if interactions among treatments occurred, or at least these findings are not reported if these tests were conducted. Moreover, I would suggest that the authors try more complex multivariate models to analyse their data; for instance, using approaches such as analysis of co-variance (ANCOVA), generalized linear models, or mixed effects models. The benefit of these more comprehensive multivariate models is that they enable the investigator to establish the relative importance of different treatments and continuous environmental variables in regulating flux.

R III: We fully agree that an experiment has to be analyzed according to its experimental design. In our case, this includes the interaction of aggregate size and soil treatment (unamended, litter addition, plant presence). We in fact have included this term in all ANOVA models, but failed to report the results when the term was not statistically significant or only weakly significant. We will fix this in the revised version. The structure of our experimental treatments is not hierarchical so that no mixed model is required. Such a model would only be necessary if one would analyse the time series data, i.e. if one had several values per microcosm. We have considered this but decided not to do so, for the following reasons:

(1) our focus was on the *average response* during distinct phases that we have identified in our time series, in particular during "hot moments" after wetting; working with average time-series data provides an answer to hypotheses about whether total emissions during this period, for example, differ between treatments; in other words, our hypotheses were about cumulated fluxes during a period, and we therefore carried out these analyses at this level.

(2) the processes we observed are extremely dynamic; fitting a full time series model would almost certainly have resulted in significant time x treatment interactions – such effects would primarily be driven by the peak values of e.g. $N_2O$ emissions after wetting; whether treatment differences for these single measurements reflect true differences in time and extent of peak fluxes is uncertain… it in fact is very likely that the true peak occurred a short time before or after these measurements, and this may be treatment specific. Again, we were not interested in whether the maximum flux occurred a bit earlier or later in time (this may not be reproducible anyways), but whether total emissions during the hot moment changed. Working with such aggregated data solves the problem of subtle shifts in emission timing, and gives extreme values much less weight.

(3) the proper modelling of the time series is very complicated: this involved heterogeneous variances (because large values scatter more) and the modelling of serial correlations (because subsequent values are not independent). On the time-aggregated scale, these problems do not occur. We also could log-transform the data to compare the treatments, which was not possible on the raw data because (a) negative values occurred due to measurement error, and (b) we were asking questions about total fluxes (e.g. grams of $N_2O$ emitted) and not relative effects.

In summary, we agree that more complex analyses can potentially be done. However, we have deliberately focused on (1) the aggregation level that matched the questions we were asking, and (2) the aggregation level at which statistical procedures were robust. We agree that we did not document this very well and propose to address this in the revision.

Third, I agree with the first referee that the authors need to spend a bit more time clearly highlighting what knowledge gaps this paper fills. As the first referee indicates, there are already existing studies that have examined the individual effects of all the variables discussed here. In order to make this paper more impactful, the authors need to articulate how this specific study is unique or advances our current state-of-knowledge (e.g. does the factorial design add knowledge or insight?). Specific comments are provided in the section below.

R IV: We concur with both reviewers that the specific objectives of this study were not sufficiently well stated. As mentioned in our response to Reviewer 1, this aspect will be addressed. We will clarify that, while the effects of microhabitats related to soil aggregates, the detritusphere and plant-soil interactions in the rhizosphere on $N_2O$ emissions from soils have been studied individually, little is known about their relative effects and interactions. In our mesocosm study, we investigated this aspect for the hot moments of $N_2O$ emissions from floodplain soils during the drying phase after flooding. In particular, aggregate size effects have not been investigated in this context (as stated on lines 79f). A particular novel aspect of the study is the minimization of the potentially confounding factor "soil structure" by mixing a given aggregate size fraction with inert material replacing the removed smaller or larger fraction. As stated on line 71ff, previous studies employing isolated aggregate size fractions have provided partially inconsistent results possibly linked to some extent to the changes in soil structure by aggregate separation.

The better specified objectives and novel aspects will be included in the introduction of the revised manuscript.

SPECIFIC COMMENTS

1. Lines 136-137: For experimental purposes, the investigators have created quasi-artificial system conditions, with treatments either containing macro- or microaggregates. While I fully understand why this was done, it would be useful to understand (even qualitatively) how close or far from reality these treatments are. For example, what was the proportion of macro- and micro-aggregates under natural conditions?

R1: The original floodplain soil consisted of 18.5 ± 4.6 % aggregates smaller than 250 μm and 81.5 ± 4.6 % macroaggregates (mean ± sd; n = 10). We composed our model soils of a 1:1 mixture of isolated aggregates and inert matrix material. This is different from the original soil composition, but well within the range of published top soil aggregate size distributions (e.g. Cantón et al., 2009; Gajić et al., 2010; Six et al., 2000). 50% microaggregates may be more than what is found in most natural or agricultural soils. Nevertheless, we chose to use equal amounts of small and large aggregates to be able so separate effects of aggregate size from effects of aggregate amount (soil mass). To reflect these reasonings, we propose to discuss the distribution of small and large aggregates in the original soil (material and method section of the revised manuscript). The discussion of relevance would be added to the discussion in section 4.1 and in the conclusions. For additional considerations on the effect of flood disturbance on small-scale heterogeneity and dynamics of aggregate size distribution see R5 below.

2. Line 173: Clarity of expression; consider revising this section to read "The mesocosm experiment had a factorial experimental design consisting of two factors (model soil and plant-soil treatment), with the first factor containing two levels (macroaggregates, microaggregates) and the second factor containing three levels (unamended, litter added, plant present). This experimental design resulted in six treatments, each replicated six times."

R2: The authors concur with this remark and will adjust this part accordingly

3. Line 179-180: What was the rationale for autoclaving the leaves? Under natural conditions, these leaves would contain their own microbial community which could contribute to $N_2O$ dynamics, and autoclaving means that the results will be biased towards the activity of the soil community (or, spore-forming phyllosphere microbes able to resist the effects of autoclaving).

R3: Since we specifically wanted to test the effect of additional labile C available to the $N_2O$ producing or consuming soil microbial community, we decided to eliminate, or at least reduce the effect of and interaction with the phyllosphere of the collected leaves by sterilization. We are aware that this introduces a certain bias. However, so far there are no direct effects of the phyllosphere community on $N_2O$ production described in the literature. The only role of these organisms in plant-atmosphere interactions reported in the literature is in capturing/consuming methane and/or volatile organic carbon compounds (Bringel and Couée, 2015). On the other hand, we cannot say anything about potential effects of interactions between the phyllosphere and soil communities on $N_2O$ production/consumption. These remarks will be added to the discussion section of the litter effects, 4.2., in the revised version of the manuscript.

4. Lines 221-232: Further detail on the statistical analyses are required here. For example, what were the independent variables used in the ANOVA? Did the model include interaction terms? Given that sampling was conducted over different periods of time, did the authors use a repeated measures ANOVA, to account for the effects of time?

R4: The independent variables for the two way ANOVA were SOIL TREATMENT (unamended, litter addition, plant presence) and AGGREGATE SIZE. The ANOVA model also included interactions, which were indeed significant for some of the parameter. However, we did not report the cases where the interaction was not or only weakly statistically significant. We will address this in the revision.
Our hypotheses were related to total fluxes during hot moments, which is why we did not analyze the time series but aggregated data. The rationale for this was already explained in detail above (R III).

5. Lines 300-353: This is an interesting and well-written part of the discussion. However, I do think that this part of the discussion could be improved by trying to link back the findings from the experiment to natural conditions (see point 1). For example, under natural conditions, what is the relative distribution of macro- or micro-aggregates? Based on your understanding/knowledge of the natural aggregate distributions, what patterns or processes do you think will dominate in a natural setting? While I realise this might be somewhat speculative (unless other data, such as field measurements, are available), I think it's an important talking point, as it will enable the reader to relate these findings (derived under somewhat artificial conditions) to the real world.

R5: For our assessment and evaluation of the relative distribution of macro- and micro-aggregates in our experimental soil and other soils reported in the literature see R1.

Furthermore, the frequent hydrological disturbance in floodplains creates a highly dynamic and small-scaled spatial mosaic of different aggregate size distributions. Therefore, the results on the individual potentials of differently sized aggregates to emit $N_2O$ and their respective interactions with plant roots and litter accumulation could help to better understand the seemingly erratic spatial and temporal distribution of enhanced $N_2O$ emissions from floodplain areas. Considering our results, one could speculate that zones with a relatively high percentage of macroaggregates would be prone to particularly high emissions during hot moments. In a revised manuscript, these considerations would be added also to the discussion in section 4.1.

6. Lines 380-406: The discussion of potential direct and indirect effects facilitated by the presence of an active root system is interesting and well-reasoned. However, I was left wondering as to how generalizable these findings are, given the wide range of traits displayed by different plants? I.e. to what extent are the trends identified here unique to Salix, and to what extent are these patterns more widely generalizable? I think it is important that the authors develop this section a bit further, in particular acknowledging this limitation more frankly.

R6: Different plant species may indeed exert different rhizosphere effects (for an overview of potential rhizosphere effects see the current manuscript lines 81 to 101). Thus, strictly speaking, this study is directly relevant only for salix sp.. However, this is an important plant genus adapted to temporary flooding and thus often found in river floodplains. While oxygen depletion by root exudation stimulated microbial respiration, discussed as one process potentially reducing $N_2O$ emissions in our study, likely occurs in the rhizosphere of any plant, rhizosphere aeration as alternative process is restricted to plants possessing aerenchyma. However, the latter is a trait of many plants adapted to temporary flooding.  It has been described also for the grass family of poaceae, or for ash, and It would not be surprising to find this trait in other Salicaceae like poplar sp. and other species of softwood floodplain forests.

[revised manuscript text omitted]

microhabitats associated with soil aggregates, the detritusphere and plant–soil interactions on $N_2O$ emissions
from floodplain soils under changing pore-space saturation. We simulated a flooding event in mesocosm
experiments with main focus on the dynamics of $N_2O$ emissions during hot moments in the drying phase after
flooding. To isolate the effect of aggregate-size and to minimize confounding effects of differences in soil
structure, we prepared model soils by mixing aggregate size fractions of a floodplain soil with suitable inert
material. The combined effects of soil aggregate size and plant detritus or plant-soil interactions were addressed
by mixing the model soils with leaf litter or by planting them with willow cuttings (*Salix viminalis* L.).

**Kommentar [MaL2]:** Ref 1, Reply; Ref 2, R IV

We demonstrate that the level of soil aggregation significantly affects $N_2O$ emission rates from floodplain soils
through its modulating control on the model soil's physicochemical properties. We further show that these
effects can be modified by the presence of a detritusphere and by root–soil interactions, changing carbon and N
substrate availability and redox conditions.

**2. Material and methods**

**2.1 Model soils**

In February 2014, material from the uppermost 20 cm of a N-rich gleyic Fluvisol (calcaric, humic siltic) with
20% sand and 18% clay (Samaritani et al., 2011) was collected in the restored Thur River floodplain near
Niederneunforn (NE Switzerland 47°35' N, 8°46' E, 453 m.a.s.l.; MAT 9.1 ℃; MAP 1015 mm). After removing
plant residues such as roots, twigs and leaves, the soil was mixed and air-dried to a gravimetric water content of
$24.7 \pm 0.4$ %. In the next step, the original floodplain soil material, consisting of $18.5 \pm 4.6$ % aggregates smaller
than 250 µm and $81.5 \pm 4.6$ % macroaggregates (mean $\pm$ SD; n = 10), was separated into a macroaggregate

**Kommentar [MaL3]:** Ref 2, R1

fraction (250–4000 µm) and a microaggregate fraction (< 250 µm) by dry sieving. The threshold of 250 µm
between macroaggregates and microaggregates was chosen based on Tisdall and Oades (1982). Soil aggregate
fractions were then used to re-compose model soils. In order to preserve soil structure, the remaining aggregate
size fractions were complemented with an inert matrix replacing the removed aggregate size fraction of the
original soil. Model Soil 1 (LA) was composed of soil macroaggregates mixed in a 1:1 (w/w) ratio with glass
beads of 150–250 µm size serving as inert matrix material replacing the microaggregates of the original soil.
Similarly, Model Soil 2 (SA) was composed of soil microaggregates mixed at the same ratio with fine quartz
gravel of 2000–3200 µm size. To generate an even mixture of original soil aggregates and the respective inert
matrix a Turbula mixer (Willy A. Bachofen AG, Muttenz, Switzerland) was used. The proportions of the
aggregate size fractions in the model soils were different from the original soil, and 50% microaggregates may
be more than what is found in most natural or agricultural soils (often less than 10 %). Nevertheless, we chose to
use equal amounts of micro- and macroaggregates, in order to be able to separate the effects of aggregate size
from effects of aggregate amount (soil mass). These proportions were still well in the range of common top soils
(e.g. Cantón et al., 2009; Gajić et al., 2010; Six et al., 2000). The physicochemical properties of the two soils

**Kommentar [MaL4]:** Ref 2; RI, R1

were determined by analysing three random samples of each model soil. Texture of the complete model soils
was determined using the pipette method (Gee and Bauder, 1986) and pH was measured potentiometrically in a
stirred slurry of 10 g soil in 20 ml of 0.01 M $CaCl_2$, as recommended in Hendershot et al. (2007). Additionally

**Kommentar [MaL5]:** Ref 1, R2

organic carbon ($C_{org}$) and total nitrogen (TN) were analysed in both aggregate size fractions without the inert material, using the method described by Walthert et al. (2010). The two model soils displayed very similar physicochemical properties (Table 1), except for the C:N ratio that was lower in macroaggregates than in microaggregates. The latter was due to the slightly lower organic carbon content in concert with slightly higher TN values in the macroaggregates. The high calcium carbonate ($CaCO_3$) content of the source material of our model soils ($390 \pm 3$ g $CaCO_3$ $kg^{-1}$; Samaritani et al., 2011) buffered the systems at an alkaline pH of $8.00 \pm 0.02$ for LA and $7.56 \pm 0.01$ for SA respectively (Table 1), ensuring that the activity of key N-transforming enzymes was not hampered by too low pH, and that the potential for simultaneous production and consumption of $N_2O$ in our experiment was fully intact (Blum et al., 2018; Frame et al., 2017).

**2.2 Mesocosms**

For the mesocosm experiments, transparent polyvinyl chloride (PVC) cylinders with polymethyl methacrylate (PMMA) couplings were used. A mesocosm comprised a bottom column section, containing the soil material and a drainage layer as described below, and the upper headspace section with a detachable headspace chamber (Fig. 1). Each column section was equipped with two suction cups (Rhizon MOM Soil Moisture Samplers, Rhizosphere Research Products, Netherlands; pore size 0.15 μm) for soil solution sampling. The suction cups were horizontally inserted at 5 cm and 20 cm below soil surface. For redox potential measurements, two custom-made Pt electrodes (tip with diameter of 1 mm and contact length of 5 mm) were placed horizontally at a 90° angle to the suction cups at the same depths, with the sensor tip being located 5 cm from the column wall. A Ag/AgCl reference electrode (B 2820, SI Analytics, Germany) was installed as shown in Fig. 1. A volumetric water content (VWC) sensor (EC-5, Decagon, USA) was installed 15 cm below the soil surface. To avoid undesired waterlogging, each column section contained a 5 cm thick drainage layer composed of quartz sand with the grain size decreasing with depth from 1 mm to 5.6 mm (Fig. 1). The upper cylinder section was equipped with three way valves for gas sampling, and an additional vent for pressure compensation.

**2.3. Experimental setup**

The mesocosm experiment had a factorial experimental design consisting of two factors (MODEL SOIL and TREATMENT), with the first factor containing two levels (macroaggregates, microaggregates) and the second factor containing three levels (unamended, litter added, plant presence). This experimental design resulted in six treatments, each replicated six times (Table 2). As basic material, each mesocosm contained 8.5 kg of either of the two model soils. Unamended model soils were used to investigate exclusively the effect of aggregate size, abbreviated as LAU (large aggregates, unamended) and SAU (small aggregates, unamended), respectively. In order to specifically assess the effect of enhanced availability of labile C in the detritusphere for the $N_2O$ producing or consuming soil microbial community, two sets of mesocosms were amended with freshly collected leaves of Basket Willow (*Salix viminalis* L.). Those leaves were cut into small pieces, autoclaved, and then added to the model soil components (8 g $kg^{-1}$ model soil) during the mixing procedure to create treatments LAL (large aggregates, litter) and SAL (small aggregates, litter), respectively. The sterilization step was included to create equal starting conditions in all litter treatments by reducing any potential effect of, and interaction with, the phyllosphere microbial community even though a direct involvement of the phyllosphere community in $N_2O$ production was unlikely according to the literature (Bringel and Couée, 2015). A third set of mesocosms was planted with cuttings collected from the same *Salix viminalis* creating treatments LAP (large aggregates, plant)

Kommentar [MaL6]: Ref. 2, R2

Kommentar [MaL7]: Ref 2, R3

and SAP (small aggregates, plant), respectively to evaluate the effects of root–soil interactions in the respective
model soils. For each mesocosm one cutting was inserted 10 cm into the soil, protruding from the surface about
3 cm.

The addition of leaf litter to the model soils led to an increase of $C_{org}$ and TN in LAL relative to LAU by 41 %
and 35 %, respectively, and in SAL relative to SAU by 58 % and 44 % respectively. The bulk density of the
unamended model soil SAU ($1.27 \pm 0.01$ g cm$^{-3}$) was slightly higher than the one of LAU ($1.22 \pm 0.01$ g cm$^{-3}$;
adj. $P$: < 0.0001). Regarding the litter addition treatments, the bulk density of LAL ($1.13 \pm 0.01$ g cm$^{-3}$) was
significantly smaller than the one of LAU (adj. $P$: < 0.0001), whereas the bulk density of SAL ($1.27 \pm 0.02$ g cm$^{-3}$) did not differ significantly from the one of SAU. The soils in the treatments with plants exhibited a similar
bulk density (LAP: $1.23 \pm 0.02$ g cm$^{-3}$; SAP: $1.24 \pm 0.01$ g cm$^{-3}$) as in the respective unamended treatments.

The experiments were conducted inside a climate chamber set to constant temperature ($20 \pm 1$ °C) and relative
air humidity ($60 \pm 10\%$), with a light/dark cycle of 14/10 h (PAR $116.2 \pm 13.7$ µmol m$^{-2}$ s$^{-1}$). The experimental
period was divided into four consecutive phases: The conditioning phase (Phase 1) lasted for 15 weeks and
allowed the model soils to equilibrate and the plants to develop a root system. This was followed by the first
experimental phase of nine days (Phase 2), serving as a reference period under steady-state conditions. During
Phases 1 and 2, the soils were continuously irrigated with artificial river water (Na$^+$: 0.43 µM; K$^+$: 0.06 µM;
Ca$^{2+}$: 1.72 µM; Mg$^{2+}$: 0.49 µM; Cl$^-$: 4.04 µM; NO$_3^-$: 0.16 µM; HCO$_3^-$: 0.5 µM; SO$_4^{2-}$: 0.11 µM; pH: 7.92) via
suction cups, to maintain a volumetric water content of $35 \pm 5$ %. In Phase 3, the mesocosms were flooded by
pumping artificial river water through the drainage vent at the bottom into the cylinder (10 mL min$^{-1}$, using a
peristaltic pump; IPC-N-24, Ismatec, Germany) until the water level was 1 cm above the soil surface. After 48 h
of flooding, the water was allowed to drain and the soil to dry for 18 days without further irrigation (Phase 4).

**2.4 Sampling and analyses**

During the entire experiment, water content and redox potential were automatically logged every 5 minutes
(EM5b, Decagon, USA and CR1000, Campbell scientific, USA, respectively).

At selected time points during the experiment, soil-emitted gas and soil solution were sampled. For N$_2$O flux
measurements, 20, 40 and 60 minutes after closing the mesocosms, headspace gas samples (20 mL) were
collected using a syringe and transferred to pre-evacuated exetainers. The samples were analyzed for their N$_2$O
concentration using a gas chromatograph (Agilent 6890, Santa Clara, USA; Porapak Q column, Ar/CH$_4$ carrier
gas, micro-ECD detector). Measured headspace N$_2$O concentrations were converted to moles using the ideal gas
law and headspace volume. The N$_2$O efflux rates were calculated as the slope of the linear regression of the N$_2$O
amounts at the three sampling times, relative to the exposed soil surface area (Fig. 1, Shrestha et al., 2012).

For soil water sampling, 20 mL of soil solution were collected using the suction cups. Water samples were
analyzed for dissolved organic carbon (DOC) and TN concentrations with an elemental analyzer (Formacs$^{HT/TN}$,
Skalar, The Netherlands). Nitrate and ammonium concentrations were measured by ion chromatography (IC 940,
Metrohm, Switzerland), and nitrite (NO$_2^-$) concentrations were determined photometrically (DR 3900, Hach
Lange, Germany).

**2.5 Data analyses**

We were interested in effects on cumulated N$_2$O emissions during hot moments following flooding. We
therefore analyzed data aggregated over this period rather than the raw full time series data. This procedure also avoided potential issues with small shifts in the timing of emissions that might have been significant but which
were irrelevant for the total fluxes we focused on. The total amount of $N_2O$ emitted during the period of

**Kommentar [MaL8]:** Ref 2, R III

enhanced $N_2O$ fluxes in Phase 4, $Q_{tot}$, was calculated by integrating the $N_2O$ fluxes between day 11 and 25 of the
experiment as follows:

$$Q_{tot} = \frac{1}{2} \sum_{n=1}^{n_{max}} [\Delta_n \times (q_n + q_{n+1})] \tag{1}$$

where $\Delta_n$ is the time period between the $n^{th}$ and the $n+1^{th}$ measurement, and $q_n$ and $q_{n+1}$ the mean flux on the $n^{th}$
and $n+1^{th}$ measurement day, respectively. "n=1" refers to day 11, and $n_{max}$ to day 25 of Phase 4. The integrated
$N_2O$ fluxes, as well as the average DOC and N-species concentrations in the soil solution during this period were
analyzed by performing two-way ANOVAs with the fixed terms TREATMENT and MODEL SOIL including their
interaction. In case of significant MODEL SOIL, TREATMENT or MODEL SOIL × TREATMENT effects, their causes
were inspected with the Tukey's honestly significant difference (HSD) post hoc test. For all data, the residuals of

**Kommentar [MaL9]:** Ref 2, R III, R4

the ANOVA models were inspected, and the Shapiro–Wilk normality test was applied to ensure that the values
follow a Gaussian distribution. In case that this requirement for ANOVA was not met, the respective data set
was log-transformed. Significance and confidence levels were set at $\alpha < 0.05$. The results of the performed
ANOVAs are summarized in Table 3. For the statistical analyses we used GraphPad Prism (GraphPad Software
Inc., 2017) and R (R Core Team, 2018).

**3. Results**

**3.1 Soil moisture and redox potential**

[revised manuscript text omitted]

Tiedje, 1988; Rabot et al., 2014; Shrestha et al., 2012). We observed that the six treatments had a substantial
effect on the magnitude and temporal pattern of $N_2O$ emissions that could only be captured by observations at
relatively high temporal resolution. The fast occurrence of strong $N_2O$ fluxes over a comparatively short period
in the litter-amended treatment on the one side, and the relatively weak response to the flooding in the plant
treatment on the other, suggests complex interactive mechanisms related to distinct microhabitat effects leading
to characteristic periods of enhanced $N_2O$ emission. Rabot et al. (2014) explained $N_2O$ emission peaks during the
desaturation phase with the release of previously produced and entrapped $N_2O$. Such a mechanism may partly
contribute to high $N_2O$ emissions in our experiment initially, but the continuing depletion of $NO_3^-$ and $NO_2^-$
during the phase of high $N_2O$ emissions indicates that the flooding and drying has strong effects on N
transformations mediated by microorganisms in the soil (e.g., the balance and overall rates of nitrification,
nitrifier–denitrification, and denitrification). Hence, physical controls alone clearly do not explain the observed
timing and extent of hot moments with regard to $N_2O$ emission. In the following sections we will discuss how
the effect of flooding on microbial $N_2O$ production is modulated by differential microhabitat formation (and
hence redox conditions) in the various treatments.

**4.1 Effect of aggregate size on $N_2O$ emissions**

LA model soils exhibited both higher peak and total $N_2O$ emissions during the hot moment in the drying phase
than SA model soils (Figs. 2 and 5). By contrast, in the presence of a growing willow, there was no detectable
effect of aggregate size on the overall $N_2O$ emission (further discussion below). The aggregate size effects
observed in the unamended and litter treatments can be explained by factors controlling (i) gas diffusion (e.g.
water film distribution, tortuosity of the intra-aggregate pore space) and (ii) decomposition of encapsulated soil
organic matter (SOM) regulating the extent of $N_2O$ formation (Neira et al., 2015). In order to isolate the effect of
aggregate size (i.e., to minimize the effect of other factors that are likely to influence gas diffusion), we created
model soils of similar soil structure and texture (see Materials and Methods). We thereby implicitly accepted that
potential interactions of the two size fractions with each other, or with soil structures larger than 4 mm could not
be assessed in this experiment. Although this approach thus represents only an approximation of real-world
conditions it was still an improvement compared to experiments where no attempts were made to conserve soil
structure. Similarly, the bulk soil chemical properties of the two aggregate size fractions, such as $C_{org}$ content
and TN, are essentially the same. Differences in the initial C:N ratio and pH, although statistically significant,
can be considered equivalent in the ecological context, e.g., in terms of organic matter degradability. Therefore,
we assume in the following that the differences in $N_2O$ emissions among the treatments can mainly be attributed
to size-related aggregate properties and their interactions with litter addition or rhizosphere effects.

[revised manuscript text omitted]

> **Kommentar [MaL15]:** Ref 1; R3

According to Fender et al. (2013), in vegetated soils, microbial respiration is stimulated by deposition of root exudates, which in concert with root respiration in a highly saturated pore space, leads to severe and ongoing oxygen depletion. Under such stable anoxic conditions complete denitrification would take place generating $N_2$ and not $N_2O$ as the dominant final product and therefore $N_2O$ emissions would be low.

While oxygen depletion by root-exudation-stimulated microbial respiration likely occurs in the rhizosphere of any plant, rhizosphere aeration is restricted to plants possessing aerenchyma. However, the latter is a characteristic of many plants adapted to temporary flooding, and has been described also for *Poaceae*, or for ash. Furthermore, it is reasonable to expect this trait to be found in other *Salicaceae* like *Populus sp.* and other species of softwood floodplain forests. In areas with monospecific stands of, for example *Salix sp.*, which are often found on restored river banks, this $N_2O$-emission reducing trait can be a welcome side effect.

> **Kommentar [MaL16]:** Ref 2; R6

**5. Conclusions**

[revised manuscript text omitted]

Butterbach-Bahl, K., Baggs, E. M., Dannenmann, M., Kiese, R. and Zechmeister-Boltenstern, S.: Nitrous oxide
emissions from soils: how well do we understand the processes and their controls?, Philos. Trans. R. Soc. Lond.
B. Biol. Sci., 368(1621), 20130122, doi:10.1098/rstb.2013.0122, 2013.

Cantón, Y., Solé-Benet, A., Asensio, C., Chamizo, S. and Puigdefábregas, J.: Aggregate stability in range sandy
loam soils Relationships with runoff and erosion, CATENA, 77(3), 192–199, doi:10.1016/j.catena.2008.12.011,
2009.

Ciais, P., Sabine, C., Bala, G., Bopp, L., Brovkin, V., Canadell, J., Chhabra, A., DeFries, R., Galloway, J.,
Heimann, M., Jones, C., Quéré, C. Le, Myneni, R. B., Piao, S. and Thornton, P.: Carbon and Other
Biogeochemical Cycles, in Climate Change 2013 - The Physical Science Basis, edited by Intergovernmental
Panel on Climate Change, pp. 465–570, Cambridge University Press, Cambridge, 2013.

Diba, F., Shimizu, M. and Hatano, R.: Effects of soil aggregate size, moisture content and fertilizer management
on nitrous oxide production in a volcanic ash soil, Soil Sci. Plant Nutr., 57(5), 733–747,
doi:10.1080/00380768.2011.604767, 2011.

Drury, C. ., Yang, X. ., Reynolds, W. . and Tan, C. .: Influence of crop rotation and aggregate size on carbon
dioxide production and denitrification, Soil Tillage Res., 79(1), 87–100, doi:10.1016/j.still.2004.03.020, 2004.

Ebrahimi, A. and Or, D.: Microbial community dynamics in soil aggregates shape biogeochemical gas fluxes
from soil profiles – upscaling an aggregate biophysical model, Glob. Chang. Biol., 22(9), 3141–3156,
doi:10.1111/gcb.13345, 2016.

Elliott, A. E. T. and Coleman, D. C.: Let the soil work for us, Ecol. Bull., (39), 22–32, 1988.

Fender, A.-C., Leuschner, C., Schützenmeister, K., Gansert, D. and Jungkunst, H. F.: Rhizosphere effects of tree
species – Large reduction of $N_2O$ emission by saplings of ash, but not of beech, in temperate forest soil, Eur. J.
Soil Biol., 54, 7–15, doi:10.1016/j.ejsobi.2012.10.010, 2013.

Forster, P., Ramaswamy, V., Artaxo, P., Berntsen, T., Betts, R., Fahey, D. W., Haywood, J., Lean, J., Lowe, D.
C., Myhre, G., Nganga, J., Prinn, R., Raga, G., Schulz, M. and Van Dorland, R.: Changes in Atmospheric
Constituents and in Radiative Forcing, in Climate Change 2007: The Physical Science Basis, edited by S.
Solomon, D. Qin, M. Manning, Z. Chen, M. Marquis, K. B. Averyt, M. Tignor, and H. L. Miller, pp. 129–234,
Cambridge University Press, Cambridge, United Kingdom and New York, NY, USA, 2007.

Frame, C. H., Lau, E., Joseph Nolan, E., Goepfert, T. J. and Lehmann, M. F.: Acidification enhances hybrid $N_2O$
production associated with aquatic ammonia-oxidizing microorganisms, Front. Microbiol., 7(JAN), 1–23,
doi:10.3389/fmicb.2016.02104, 2017.

Gajić, B., Đurović, N. and Dugalić, G.: Composition and stability of soil aggregates in Fluvisols under forest,
meadows, and 100 years of conventional tillage, J. Plant Nutr. Soil Sci., 173(4), 502–509,
doi:10.1002/jpln.200700368, 2010.

Gee, G. W. and Bauder, J. W.: Particle-size Analysis, in Physical and Mineralogical Methods-Agronomy
Monograph no. 9, edited by A. Klute, pp. 383–411, American Society of Agronomy-Soil Science Society of
America, Madison, WI., 1986.

Goldberg, S. D., Knorr, K. H., Blodau, C., Lischeid, G. and Gebauer, G.: Impact of altering the water table
height of an acidic fen on $N_2O$ and NO fluxes and soil concentrations, Glob. Chang. Biol., 16(1), 220–233,
doi:10.1111/j.1365-2486.2009.02015.x, 2010.

GraphPad Software Inc.: GraphPad Prism 7.04, La Jolla, CA, www.graphpad.com, 2017.

Groffman, P. M. and Tiedje, J. M.: Denitrification Hysteresis During Wetting and Drying Cycles in Soil, Soil Sci.
Soc. Am. J., 52(6), 1626, doi:10.2136/sssaj1988.03615995005200060022x, 1988.

Hartmann, D. J., Klein Tank, A. M. G., Rusticucci, M., Alexander, L. V, Brönnimann, S., Charabi, Y. A.-R.,
Dentener, F. J., Dlugokencky, E. J., Easterling, D. R., Kaplan, A., Soden, B. J., Thorne, P. W., Wild, M. and
Zhai, P.: Observations: Atmosphere and Surface, in Climate Change 2013 - The Physical Science Basis, edited
by Intergovernmental Panel on Climate Change, pp. 159–254, Cambridge University Press, Cambridge., 2013.

Hefting, M., Clément, J.-C., Dowrick, D., Cosandey, A. C., Bernal, S., Cimpian, C., Tatur, A., Burt, T. P. and
Pinay, G.: Water table elevation controls on soil nitrogen cycling in riparian wetlands along a European climatic
gradient, Biogeochemistry, 67(1), 113–134, doi:10.1023/B:BIOG.0000015320.69868.33, 2004.

Heincke, M. and Kaupenjohann, M.: Effects of soil solution on the dynamics of $N_2O$ emissions: a review, Nutr.
Cycl. Agroecosystems, 55(2), 133–157, doi:10.1023/A:1009842011599, 1999.

Hendershot, W. H., Lalande, H. and Duquette, M.: Soil Reaction and Exchangeable Acidity, in Soil Sampling
and Methods of Analysis, edited by M. R. Carter and E. G. Gregorich, pp. 173–178, Crc Press Inc, Boca Raton,
FL., 2007.

[revised manuscript text omitted]

Totsche, K. U., Amelung, W., Gerzabek, M. H., Guggenberger, G., Klumpp, E., Knief, C., Lehndorff, E.,

Mikutta, R., Peth, S., Prechtel, A., Ray, N. and Kögel-Knabner, I.: Microaggregates in soils, J. Plant Nutr. Soil
Sci., 1–33, doi:10.1002/jpln.201600451, 2017.

Vieten, B., Conen, F., Neftel, A. and Alewell, C.: Respiration of nitrous oxide in suboxic soil, Eur. J. Soil Sci.,
60(3), 332–337, doi:10.1111/j.1365-2389.2009.01125.x, 2009.

Walthert, L., Graf, U., Kammer, A., Luster, J., Pezzotta, D., Zimmermann, S. and Hagedorn, F.: Determination
of organic and inorganic carbon, $\delta^{13}C$, and nitrogen in soils containing carbonates after acid fumigation with HCl,
J. Plant Nutr. Soil Sci., 173(2), 207–216, doi:10.1002/jpln.200900158, 2010.

Young, I. . and Ritz, K.: Tillage, habitat space and function of soil microbes, Soil Tillage Res., 53(3–4), 201–213,
doi:10.1016/S0167-1987(99)00106-3, 2000.

Zhu, X., Burger, M., Doane, T. a and Horwath, W. R.: Ammonia oxidation pathways and nitrifier denitrification
are significant sources of $N_2O$ and NO under low oxygen availability, Pnas, 110(16), 6328–6333,
doi:10.1073/pnas.1219993110/-/DCSupplemental.www.pnas.org/cgi/doi/10.1073/pnas.1219993110, 2013.

**Table 1: Physicochemical properties of the two aggregate size fractions (macroaggregates and microaggregates) and**
**added leaf litter. $C_{org}$ and TN of the aggregates were measured in triplicates. The leaf litter was analyzed in**
**quadruplicates. Final pH and texture of model soil 1 and 2 were measured in duplicates (means ± SD). Significant**
**differences in the t-tests ($P < 0.05$) are highlighted in bold.**

| | | Macroaggregates | Microaggregates | Macroaggregates vs. Microaggregates | Litter (*Salix v.* L.) |
|---|---|---|---|---|---|
| $C_{org}$ | g kg$^{-1}$ | 19.22 ± 0.55 | 21.56 ± 2.39 | P = 0.229 | 459.9 ± 2.55 |
| Total N | g kg$^{-1}$ | 1.58 ± 0.02 | 1.35 ± 0.14 | P = 0.106 | 27.39 ± 0.15 |
| C:N ratio | | 12.16 ± 0.22 | 15.99 ± 0.71 | **P = 0.007** | 16.79 ± 0.06 |
| | | Model soil 1 | Model soil 2 | Model soil 1 vs. Model soil 2 | |
| pH (CaCl$_2$) | | 8 ± 0.02 | 7.56 ± 0.01 | **P = 0.009** | |
| sand | % | 71.25 ± 0.05 | 70.7 ± 0.50 | P = 0.469 | |
| silt | % | 20 ± 0.30 | 21.1 ± 0.60 | P = 0.285 | |
| clay | % | 8.75 ± 0.25 | 8.2 ± 0.10 | P = 0.240 | |

**Table 2: Overview of treatments in the flooding–drying experiment. Model Soil 1, containing soil macroaggregates is**
**abbreviated LA, whereas Model Soil 2 contains soil microaggregates and is abbreviated SA. The last character of each**
**abbreviation stands for unamended (U), litter addition (L) and plant presence (P). Each treatment was replicated six**
**times.**

| | LAU | SAU | LAL | SAL | LAP | SAP |
|---|---|---|---|---|---|---|
| Model Soil 1 (LA) | + | - | + | - | + | - |
| Model Soil 2 (SA) | - | + | - | + | - | + |
| Leaf litter (*Salix v.*) | - | - | + | + | - | - |
| *Salix v.* | - | - | - | - | + | + |

**Table 3: Results of the two-way analysis of variance (ANOVA) of the integrated fluxes ($Q_{tot}$) and the mean**
**concentrations of chemical properties in soil solution (n=6) during the period of enhanced N$_2$O emissions (from day 11**
**to day 25). Shown are *P* values with significant differences ($P < 0.05$) highlighted in bold characters.**

| | $Q_{tot}$ | DOC | $NO_3^-$ | $NO_2^-$ | $NH_4^+$ |
|---|---|---|---|---|---|
| TREATMENT | **0.0003** | **0.0133** | 0.0988 | **< 0.0001** | **0.0007** |
| MODEL SOIL | **0.0002** | **< 0.0001** | 0.2181 | **< 0.0001** | **0.0004** |
| TREATMENT × MODEL SOIL | **0.0145** | **< 0.0001** | 0.0668 | 0.1174 | **< 0.0001** |

**Figure Captions**

**Figure 1: Schematic of a mesocosm with gas sampling valves (1), Ag/AgCl reference electrode (2), Pt redox electrodes**
**(3), suction cups (4), volumetric water content sensors (5), vent (6), and water inlet/outlet (7). The top part is only**
**attached during gas sampling.**

**Figure 2: Mean $N_2O$ emission during the flooding–drying experiment from large-aggregate model soil (LA; filled**
**circles) and small-aggregate model soil (SA, open circles). The corresponding water-filled pore space (WFPS) in LA**
**(filled triangles) and SA (open triangles) are depicted on the right Y-axis. Unamended soils (A), litter addition (B) and**
**plant treatment (C). Flooding phase indicated by the grey area. Symbols indicate means; error bars are SE; n= 6.**

**Figure 3: Redox potential relative to standard hydrogen electrode during the flooding–drying experiment in 5 cm and**
**20 cm depth (mean ± SE; n=6). Unamended soils (a and d, respectively), litter addition (b and e, respectively), plant**
**treatment (c and f, respectively). LA (filled circles) and SA (open circles); the dotted line at 250 mV marks the**
**threshold, below which denitrification is expected to occur.**

**Figure 4: DOC (circles), nitrate (squares), nitrite (diamonds) and ammonium (triangles) concentrations in pore water**
**during the flooding–drying experiment. LA (filled symbols) and SA (empty symbols). Unamended soils (a, d, g and j,**
**respectively), litter addition (b, e, h and k, respectively) and plant treatment (c, f, j and l, respectively).; (mean ± SE;**
**n=6).**

**Figure 5: Integrated $N_2O$ fluxes over the 14 days period of elevated $N_2O$ emissions in the drying phase of the flooding–**
**drying experiment (mean ± SE; n= 6). Black bars represent Model Soil 1 (macroaggregates 250-4000µm) whereas**
**Model Soil 2 (microaggregates < 250µm) is depicted as white bars. Significant differences among the six treatments**
**are denoted by different lower case letters at adj. P < 0.05.**

[Figure]

Figure 1

[Figure]

Figure 2

[Figure]

Figure 3

[Figure]

Figure 4

[Figure]

Figure 5